# Plasmonic trimers designed as SERS-active chemical traps for subtyping of lung tumors

Xing Zhao [1,5], Xiaojing Liu[2,5], Dexiang Chen [1,5], Guodong Shi[3], Guoqun Li [1], Xiao Tang [1], Xiangnan Zhu[1], Mingze Li [1], Lei Yao [1], Yunjia Wei[1], Wenzhe Song [1], Zixuan Sun [1], Xingce Fan [1], Zhixin Zhou[4], Teng Qiu [1] ✉ & Qi Hao [1] ✉

Plasmonic materials can generate strong electromagnetic fields to boost the Raman scattering of surrounding molecules, known as surface-enhanced Raman scattering. However, these electromagnetic fields are heterogeneous, with only molecules located at the 'hotspots', which account for ≈ 1% of the surface area, experiencing efficient enhancement. Herein, we propose patterned plasmonic trimers, consisting of a pair of plasmonic dimers at the bilateral sides and a trap particle positioned in between, to address this challenge. The trimer configuration selectively directs probe molecules to the central traps where 'hotspots' are located through chemical affinity, ensuring a precise spatial overlap between the probes and the location of maximum field enhancement. We investigate the Raman enhancement of the $Au@Al_2O_3$-Au-$Au@Al_2O_3$ trimers, achieving a detection limit of $10^{-14}$ M of 4-methylbenzenethiol, 4-mercaptopyridine, and 4-aminothiophenol. Moreover, single-molecule SERS sensitivity is demonstrated by a bi-analyte method. Benefiting from this sensitivity, our approach is employed for the early detection of lung tumors using fresh tissues. Our findings suggest that this approach is sensitive to adenocarcinoma but not to squamous carcinoma or benign cases, offering insights into the differentiation between lung tumor subtypes.

Surface-enhanced Raman scattering (SERS) employs plasmonic nanomaterials to generate localized intense electromagnetic fields, namely 'hotspots', around the materials to amplify the Raman scattering of nearby molecules, and has found widespread applications in sensing[1–4], analytical chemistry[5,6], catalysis[7–9] and other fields[10–12]. The 'hotspots' significantly enhance the excitation efficiency of the molecules and increase their radiation transition rate, resulting in an electromagnetic enhancement up to $10^{10}$ folds[13], and enabling highly sensitive molecular detection down to the single-molecule level[14–16].

However, these 'hotspots', which account for only less than 1% of the surface area[13,17], are highly heterogeneous, leading to the results

that only the molecules that are located at the 'hotspots' position contribute to the majority of observed SERS signal. This heterogeneity inherently restricts the sensitivity of SERS in practical sensing applications because the amount of detectable molecules among the total is actually very small[2,18]. To overcome this limitation, it is essential to ensure that the target molecules and 'hotspots' are overlapped at the same position. However, this is challenging because the construction of 'hotspots' typically employs bottom-up self-assembly methods[19–22], which often involve the use of surfactants that may lead to extra interference to the SERS signal, and can block the access of target molecules to the 'hotspots'.

[1]Key Laboratory of Quantum Materials and Devices of Ministry of Education, School of Physics, Southeast University, Nanjing 211189, PR China. [2]Department of Respiratory and Critical Care Medicine, the Affiliated Hospital of Qingdao University, Qingdao 266003, PR China. [3]Department of Thoracic Surgery, the Affiliated Hospital of Qingdao University, Qingdao 266003, PR China. [4]School of Chemistry and Chemical Engineering, Southeast University, Nanjing 211189, PR China. [5]These authors contributed equally: Xing Zhao, Xiaojing Liu, Dexiang Chen. ✉e-mail: tqiu@seu.edu.cn; qihao@seu.edu.cn

The solutions to address this limitation can be broadly classified into two categories. The first category includes strategies that employ techniques such as tip-enhanced Raman scattering (TERS)[23–27], nanocavity-enhanced Raman scattering[28–31], and shell-isolated nanoparticle enhanced Raman scattering (SHINERS)[7,32,33] to create 'hotspots' at the location of the target molecules. These approaches feature a spatial resolution down to the single-molecule level, offering a proper platform to explore the fundamental physics behind SERS. However, these approaches are generally not practical for molecular sensing applications, particularly in quantitative analysis. The second category of solutions involves introducing molecules to the 'hotspots' using chemical linkers[34,35], molecular cages[36,37] or DNA molecules[38–41]. The linkers can filter classes of small molecules into the active 'hotspots' with controlled gap distance. These techniques have facilitated the detection of specific biochemical molecules like neurotransmitters[42], but the limited types of chemical bonds in the linkers also limit the selection of guest Raman probes.

Here, we propose plasmonic Au@Al$_2$O$_3$-Au-Au@Al$_2$O$_3$ trimers, which are composed of a pair of dimer particles and a trap particle in between, to selective induce target molecules into the traps where 'hotspots' are located. The fabricated trimers were patterned over the substrate with oriented directions over square centimeter areas,

enabling Raman analysis of the sample by rapid mapping. The trimer configuration results in a substantial increase in the proportion of detectable molecules and subsequent improvement in SERS sensitivity down to the single-molecule level. The sensitivity enables in-depth investigations into pathological diagnosis of lung tumors, including the potential for tumor subtyping through SERS and exploring the correlation between the tumor node metastasis (TNM) stage and SERS outcomes.

## Results

### Preparations and characterizations of the trimer arrays

Figure 1a, b illustrates the working principles of the plasmonic trimers. For general plasmonic structures (Fig. 1a), the 'hotspots', strong SERS occurs at the gap position between two plasmonic nanoparticles, whereas the molecules are randomly adsorbed on the surface, leading to the fact that most of the molecules cannot be detected in this configuration. While for the plasmonic heterogeneous trimers (Fig. 1b), the probe molecules were mostly adsorbed on the trap particle due to the differences in chemical affinity. Therefore, the probability that a molecule can experience the amplification from the 'hotspots' is increased, leading to higher SERS sensitivity.

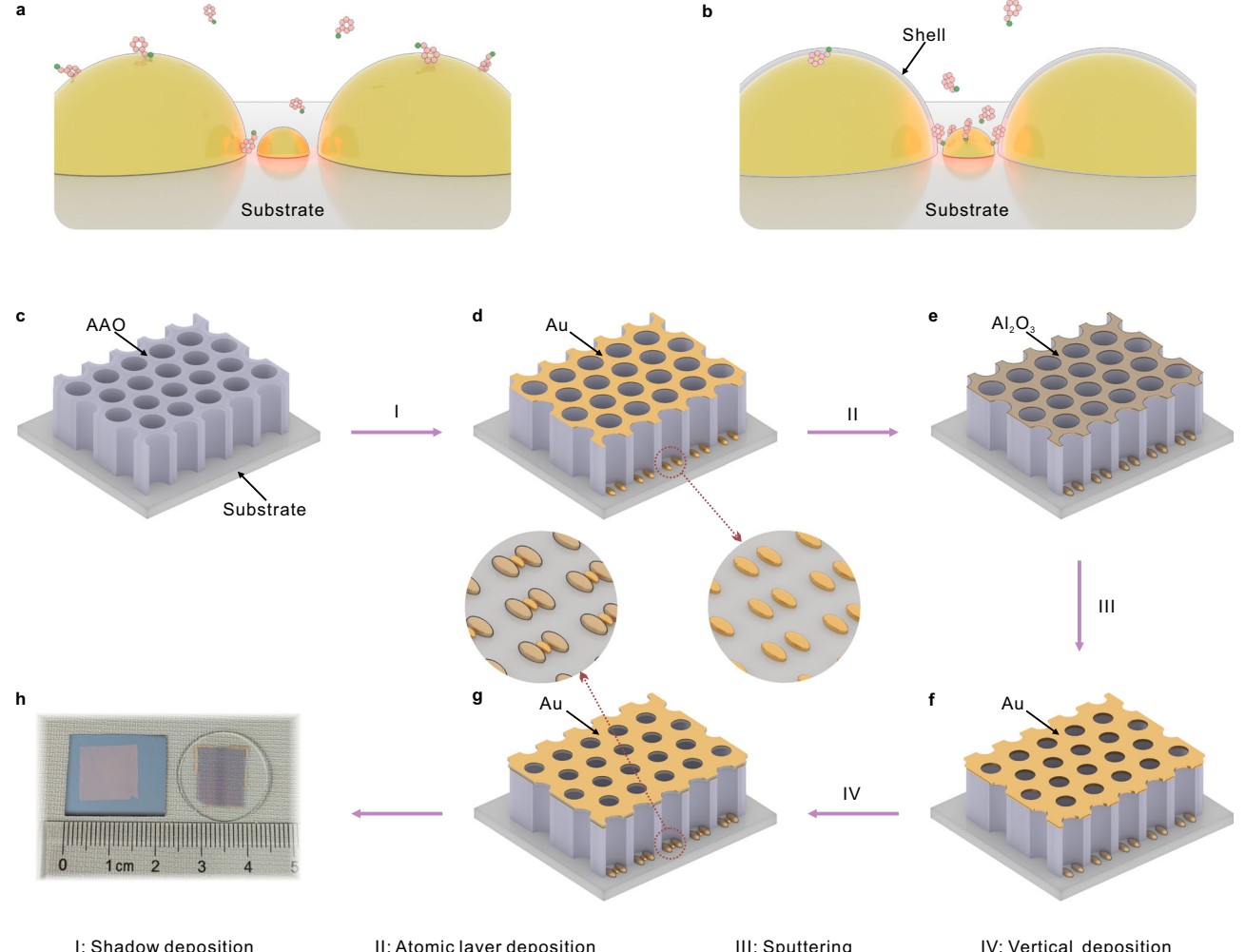

I: Shadow deposition    II: Atomic layer deposition    III: Sputtering    IV: Vertical deposition

**Fig. 1 | Schematic diagrams illustrating the preparations of the plasmonic trimer arrays.** Differences in working principles between the homogeneous trimers (**a**) and heterogeneous trimers (**b**). **c** porous AAO nanomask transferred on a substrate. **d** Fabrication of the plasmonic dimers by angle-resolved shadow depositions. **e** optional atomic layer deposition coating after the shadow depositions. **f** Further modulation of the pore diameter of AAO by sputtering. **g** Deposition of the trap particle by final vertical e-beam deposition. **h** Optical photos of the fabricated samples on silicon wafer and quartz substrates. The highlighted areas in (**a**) and (**b**) represent the "hotspots" region.

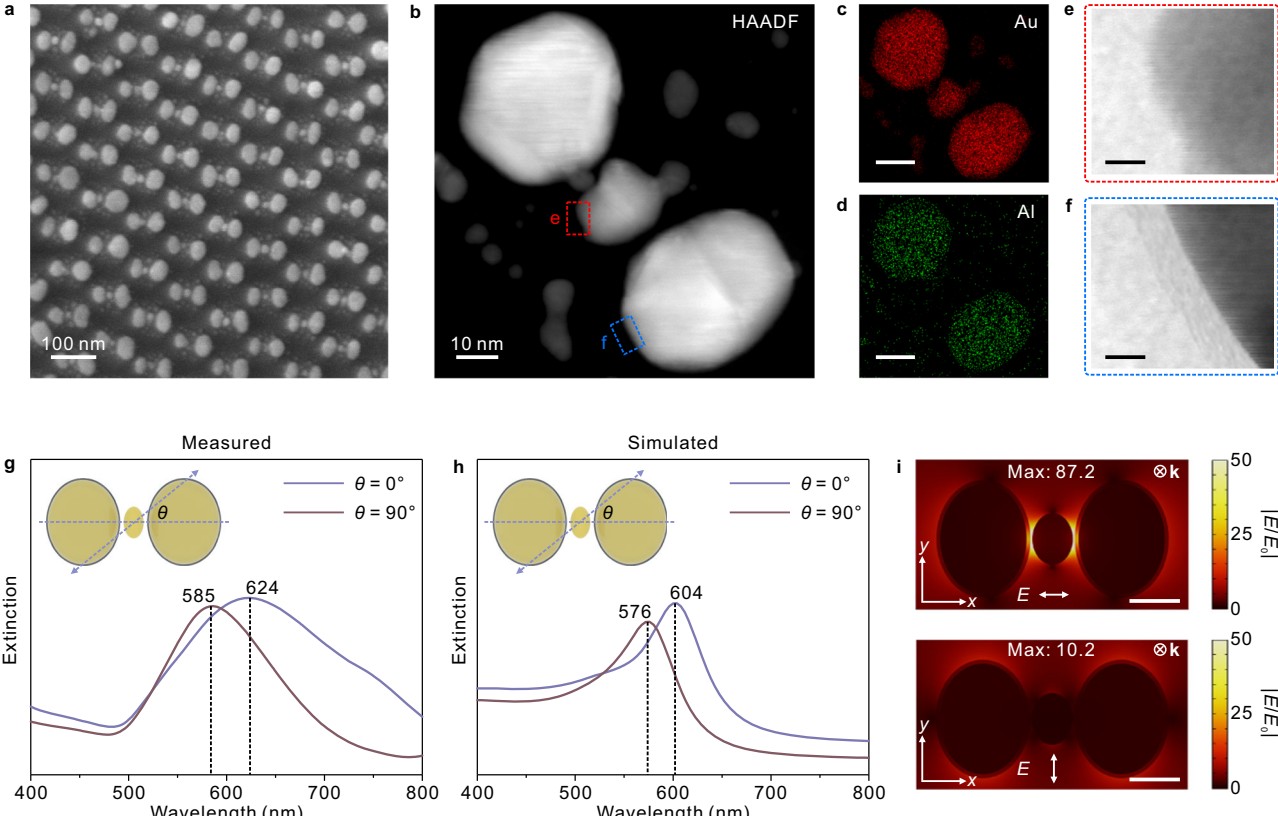

**Fig. 2 | Characterizations of the Au@Al$_2$O$_3$-Au-Au@Al$_2$O$_3$ trimers. a** SEM image of the Au@Al$_2$O$_3$-Au-Au@Al$_2$O$_3$ trimers. HAADF-STEM image (**b**) and corresponding Au (**c**) and Al (**d**) elemental mappings of the Au@Al$_2$O$_3$-Au-Au@Al$_2$O$_3$ trimers, along with TEM images for the areas marked in red (**e**) and blue (**f**). The scale bars of the elemental mapping and TEM images are 20 nm and 2 nm, respectively. Experimental (**g**) and simulated (**h**) polarized extinction spectra (horizontal: blue; perpendicular: bronze) of the Au@Al$_2$O$_3$-Au-Au@Al$_2$O$_3$ trimer arrays, respectively. **i** Local EM field distributions around the Au@Al$_2$O$_3$-Au-Au@Al$_2$O$_3$ trimers under parallel and perpendicular polarized excitations at 785 nm, and color scale corresponding to the change in electric field intensity. The scale bars are 20 nm. The SEM characterizations were independently repeated for dozens of times for samples from different batches, and the results were similar. The HAADF, elemental mappings and TEM were independently repeated for three times for samples from different batches, and the results were similar. Source data are provided as a Source Data file.

Figure 1c–h outlines the preparation of trimer arrays. The porous anodic aluminum oxide (AAO) membranes with a membrane thickness of $150 \pm 10$ nm and pore diameter of $80 \pm 2$ nm were prepared and transferred onto substrates as masks for e-beam evaporations (Fig. 1c). The methodologies describing the fabrication and manipulation of the AAO membranes with desired membrane thickness and pore diameter are detailed in refs. 43,44. The dimer particles with a gap distance of ≈ 20 nm were firstly fabricated by two-step angle-resolved shadow depositions (Fig. 1d). The deposition angles were controlled by tilting the samples to the desired angle, calculated based on the pore thickness and diameter. The gap distance between the dimers can be continuously modulated by adjusting the deposition angles (Supplementary Fig. 1). This is followed by optional atomic layer deposition (ALD) of 1.5 nm thick Al$_2$O$_3$ to form a dielectric layer on the surface on the dimers (Fig. 1e). The diameter of the AAO pores was further adjusted by magnetron sputtering (Fig. 1f), employing the same sputtering material as that will be used in the following vertical deposition. Different to e-beam evaporation, the sputtering gasified the target atoms with no directionality, and therefore, the sputtering material was mostly deposited onto the top surface of the AAO. The sputtering offers the possibility to deposit the trap particle at the gap region between the dimers in the following vertical evaporation step (Fig. 1g), and the diameter of the trap particle can be well tuned by adjusting the sputtering parameters (Supplementary Fig. 2). Afterwards, the AAO membranes were peeled off with adhesive tape, and the plasmonic

trimer arrays were exposed. The prepared sample is shown in Fig. 1h, and the dimension and shape of the sample area can be controlled by cutting the AAO membranes to a suitable size. This strategy enables large-area and mass fabrication of functional trimers with aligned orientations, small gap distances, and freedom from surface chemicals, all of which are favorable for SERS studies.

This fabrication strategy allows on-demand construction of trimer structures. The materials of both the dimer particle and trap particle can be metals or oxides, and the optional ALD process can be used to further adjust the functionality. To demonstrate the universality, we have prepared different trimers including homogeneous Au-Au-Au trimers and heterogeneous Au@Al$_2$O$_3$-Au-Au@Al$_2$O$_3$, Au-Ag-Au, and Au-TiO$_x$-Au trimers (Supplementary Fig. 3). In particular, we employed the Au@Al$_2$O$_3$-Au-Au@Al$_2$O$_3$ trimers as a typical model to illustrate how to improve the SERS performance by inducing the target molecules to the trap particle.

Figure 2a presents the scanning electron microscopy (SEM) image of the Au@Al$_2$O$_3$-Au-Au@Al$_2$O$_3$ trimers (for large-area SEM image, see Supplementary Fig. 4), suggesting that Au is deposited at the gap region between the Au@Al$_2$O$_3$ dimers. This is further identified by the high-angle annular dark-field scanning transmission electron microscopy (HAADF-STEM) in Fig. 2b. The elemental mappings in Fig. 2c, d unveil that Al only exists around the dimers. The transmission electron microscopy (TEM) images in Fig. 2e, f reveal that the ellipsoidal dimer particles were coated with an Al$_2$O$_3$ layer with a thickness of ≈ 1.5 nm, whereas the trap particle was not.

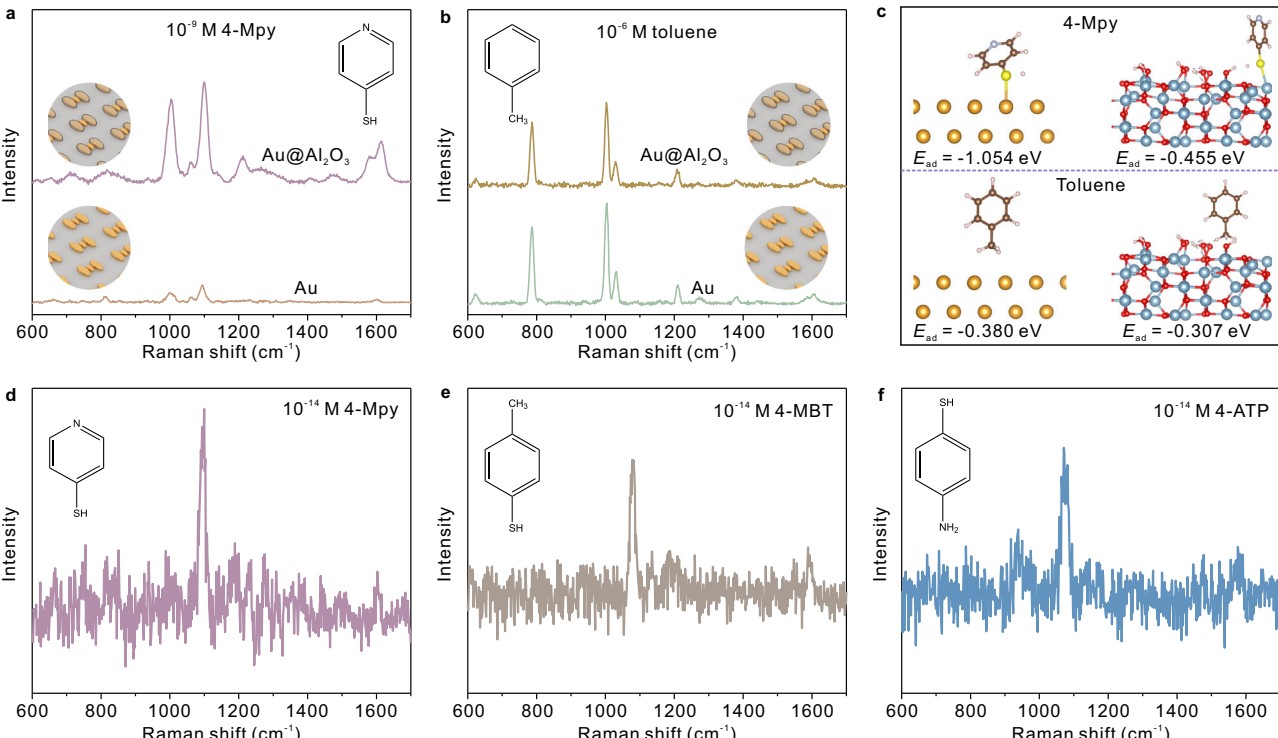

**Fig. 3 | SERS performance of the Au@Al$_2$O$_3$-Au-Au@Al$_2$O$_3$ trimer arrays. a** SERS spectra of $10^{-9}$ M 4-Mpy from the Au@Al$_2$O$_3$-Au-Au@Al$_2$O$_3$ and Au-Au-Au trimers. **b** SERS spectra of $10^{-6}$ M toluene from the Au@Al$_2$O$_3$-Au-Au@Al$_2$O$_3$ and Au-Au-Au trimers. **c** Calculated adsorption energy of 4-Mpy and toluene, respectively, on Au(111) and γ-Al$_2$O$_3$(100) surfaces. Au, Al and O atoms are represented in gold, gray and red colors. **d–f** SERS spectra of $10^{-14}$ M 4-Mpy, $10^{-14}$ M 4-MBT and $10^{-14}$ M 4-ATP from the Au@Al$_2$O$_3$-Au-Au@Al$_2$O$_3$ trimer arrays, respectively (integral time 10 s). Source data are provided as a Source Data file.

The plasmonic modes of the trimer arrays were characterized by polarized UV-vis spectroscopy. The results in Fig. 2g suggest that the modes corresponding to the long axis of the Au@Al$_2$O$_3$ ellipsoidal particles were located at approximately 585 nm under perpendicular excitation ($\theta = 90°$), and the modes of the trap particle were immersed in this case. On the other hand, a broadening and redshift in the spectrum were observed under horizontal polarization ($\theta = 0°$), due to the hybridized coupling of different modes of the trimers. The observations are basically consistent with the simulations by finite-element method (FEM) as shown in Fig. 2h. The narrowing of spectra in simulations can be explained by the differences of surface roughness and uniformity in particle size. In the simulations, a shoulder peak at 540 nm, which probably arises from the short axis mode of the Au@Al$_2$O$_3$ ellipsoidal particle, was observed under horizontal excitation ($\theta = 0°$). The assignment was confirmed by evaluating the experimental and simulated plasmonic spectra of the Au-Au-Au trimers, where the Au@Al$_2$O$_3$ dimers were replaced by bare Au dimers (see Supplementary Fig. 5).

Figure 2i plots the electromagnetic (EM) fields around the Au@Al$_2$O$_3$-Au-Au@Al$_2$O$_3$ trimers calculated by FEM simulations. The results reveal that the 'hotspots' are located at the regions around the trap particle. Obvious polarization-dependent enhancement was observed with a maximum of $E/E_0 = 87.2$ under horizontal excitation. Considering that the enhancement factor can be approximated by the fourth power of $E/E_0$, we can estimate that the average EM enhancement around the trimer structures is on the order of $10^5$ through simple calculation[45] (maximum $5.8 \times 10^7$), representing a $10^2$-fold increase compared with monomer arrays[46,47].

**SERS performance and single-molecule detection**

Figure 3 illustrates the SERS performance of the Au@Al$_2$O$_3$-Au-Au@Al$_2$O$_3$ trimers. The Au-Au-Au trimers, which were similar to

Au@Al$_2$O$_3$-Au-Au@Al$_2$O$_3$ in configuration but without the Al$_2$O$_3$ coating, were prepared for comparison. 4-Mercaptopyridine (4-Mpy), which strongly adsorbs on gold by the thiol group, was employed as a Raman probe to illustrate the chemical selectivity. As shown in Fig. 3a, the SERS signal was significantly improved on the Au@Al$_2$O$_3$-Au-Au@Al$_2$O$_3$ trimers. The improvement would be more significant with lower molecular concentrations (see Supplementary Fig. 6). It can be explained by the differences in chemical affinities between the molecules and different substrate materials. For the Au@Al$_2$O$_3$-Au-Au@Al$_2$O$_3$ trimers, the 4-Mpy molecules primarily adsorbs on the Au traps, ensuring the overlapping between the Raman probes and 'hotspots' in position, and thus substantially improve the SERS sensitivity. While for the Au-Au-Au trimers, 4-Mpy random adsorbs on the trimers, and the probability that a molecule is located at the hotspots is decreased. Additionally, comparative Raman mappings of $10^{-11}$ M of 4-Mpy were performed to illustrate the differences in SERS performance between the two trimer configurations, as evidenced in Supplementary Fig. 7. The impact of ALD thickness on SERS performance was also discussed in Supplementary Fig. 8.

As a contrast, toluene, which is structurally similar to 4-Mpy but without the thiol group, was investigated for comparison. The results in Fig. 3b suggest that there are no distinct differences in SERS intensities between the two cases. This is because toluene adsorbs on gold and Al$_2$O$_3$ with similar binding affinities via a methyl group. It should be noted that the preparations for toluene and 4-Mpy were different due to their differences in binding affinities, and the real differences in SERS sensitivity would be much more significant (see Methods section, Raman preparations). Besides, Raman characterizations of 4-Mpy and toluene at different concentrations are available in Supplementary Fig. 6.

The adsorption energies of 4-Mpy and toluene on Au(111) and γ-Al$_2$O$_3$(100) were calculated by density functional theory (DFT)

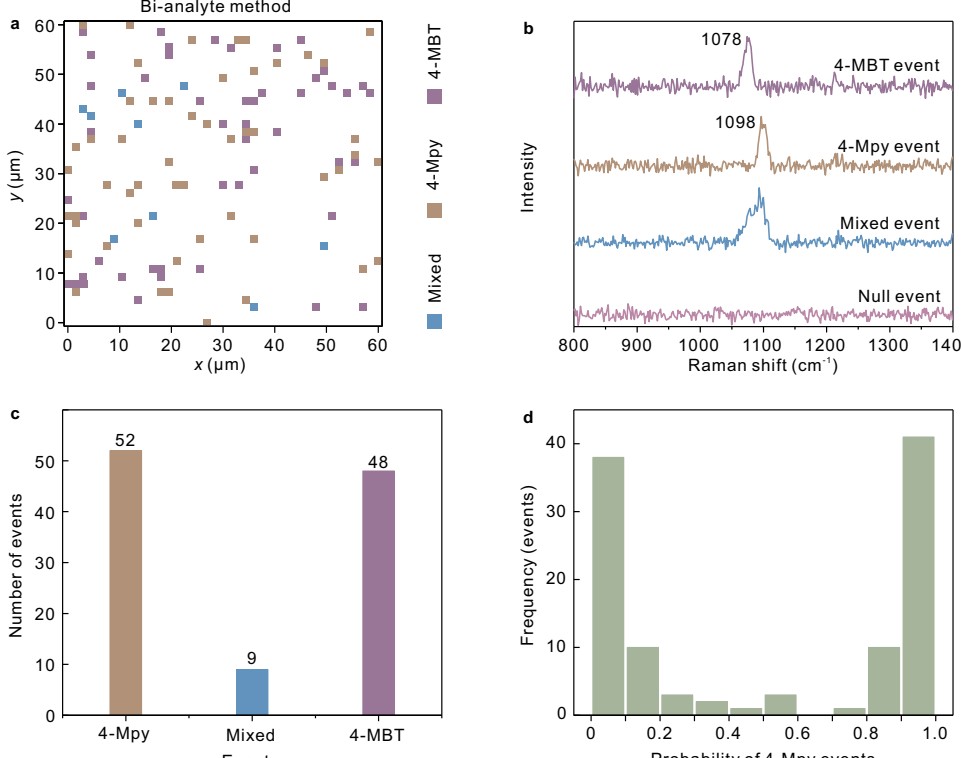

**Fig. 4 | Bi-analyte single-molecule SERS investigations with the Au@Al₂O₃-Au-Au@Al₂O₃ trimers. a** 2D statistical map of the single-molecule 4-MBT events, 4-Mpy events and mixed events by the bi-analyte method (40 × 41, step 1.5 µm, integral time 500 ms). **b** Typical SERS spectra of single-molecule 4-MBT events (purple), single-molecule 4-Mpy events (brown), mixed events (blue) and null events (pink). **c** Statistical histograms of pure 4-Mpy events (brown), mixed events (bule) and pure 4-MBT events (purple). **d** Probability histogram of the pure 4-Mpy events obtained by MPCA algorithm. Source data are provided as a Source Data file.

simulations, as depicted in Fig. 3c. The gold nanoparticles fabricated by e-beam evaporation preferentially expose Au(111) facets[48], which is confirmed by the X-ray diffraction (XRD) and high-resolution TEM results in Supplementary Fig. 9. On the other hand, γ-Al₂O₃(100) was employed for the simulation due to its similarity to the amorphous and partially hydroxylated alumina produced in the ALD process[49–51]. Details are available in the Methods section, and the full simulation results are available in Supplementary Fig. 10.

The simulations reveal a strong binding affinity between 4-Mpy and Au(111) of −1.054 eV, while a significantly weaker binding affinity was observed for γ-Al₂O₃ (−0.455 eV). The results suggest that 4-Mpy molecules would be preferably induced to the Au trap particle in the Au@Al₂O₃-Au-Au@Al₂O₃ trimer configuration, while they would be randomly adsorbed on the surface of the Au-Au-Au trimers, illustrating the differences in SERS intensities in Fig. 3a. In comparison, the binding affinities between toluene and Au(111) and γ-Al₂O₃(100) are −0.380 eV and −0.307 eV, respectively, indicating that toluene weakly adsorbs on Au(111) and Al₂O₃ by physical adsorption with similar bonding energies. Consequently, no distinct differences in the SERS intensities in Fig. 3b is observed for toluene. Additionally, the plasmonic modes of the Au@Al₂O₃-Au-Au@Al₂O₃ trimers are relative closer to the excitation laser wavelength at 785 nm (Fig. 2 and Supplementary Fig. 5), which might be a secondary reason for the differences in SERS intensities in Fig. 3a.

Figure 3d–f presents the SERS detection limit spectra of 4-Mpy, 4-methylbenzenethiol (4-MBT) and 4-aminothiophenol (4-ATP) at a concentration of $10^{-14}$ M on the Au@Al₂O₃-Au-Au@Al₂O₃ trimers, whereas their detection limits on the Au-Au-Au trimers were determined to be $10^{-11}$ M (full data in Supplementary Figs. 6, 11). Additionally, improved signal homogeneity was observed for the Au@Al₂O₃-Au-Au@Al₂O₃ trimers, suggesting the efficiency against the

heterogeneous distribution of 'hotspots' (Supplementary Fig. 12). The DFT simulations of different adsorption sites for these molecules are available in Supplementary Fig. 10. The experimental enhancement factor was calculated to be ≈ $10^{10}$ for 4-Mpy (see Methods section, calculation of SERS enhancement factors). Moreover, similar comparative SERS measurements of adenine, thiram, and benzeneselenol (BSe), which have a high affinity towards the gold trap particles, have also been performed to validate the functionality of the Au@Al₂O₃-Au-Au@Al₂O₃ trimers, as included in Supplementary Fig. 13.

Figure 4 illustrates the single-molecule SERS detection measurements on the Au@Al₂O₃-Au-Au@Al₂O₃ trimer arrays by the bi-analyted method. This method employs two analytes to sort out single-molecule SERS events by judging whether the event contains an individual analyte or two analytes[52], and the details about the statistical method in this experiment are available in ref.[53]. We selected 4-MBT and 4-Mpy, which are similar in chemical structures, as the Raman probes. The concentrations were set to be $10^{-13}$ M, and the Raman intensities at 1078 cm⁻¹ (4-MBT) and 1098 cm⁻¹ (4-Mpy) were analyzed to evaluate the SERS performance. Figure 4a plots the statistical map from 1640 measurement points, where pure 4-MBT events, pure 4-Mpy events and mixed events are marked with different colors. Here most of the events were blank due to the low molecular concentrations, which is favorable to improve the validity of single-molecule SERS[53,54]. Fig. 4b presents the typical SERS spectra of single-molecule 4-MBT events, single-molecule 4-Mpy events, mixed events and null events. Figure 4c records the statistic results from the map, including 52 pure 4-Mpy events, 48 pure 4-MBT events, and 9 mix events possessing Raman characters from both molecules. Rigorous statistical histogram of the single-molecule frequency by modified principal-component analysis (MPCA) algorithm was plotted in Fig. 4d (details see Supplementary Fig. 14 and Methods section, computational steps for MPCA

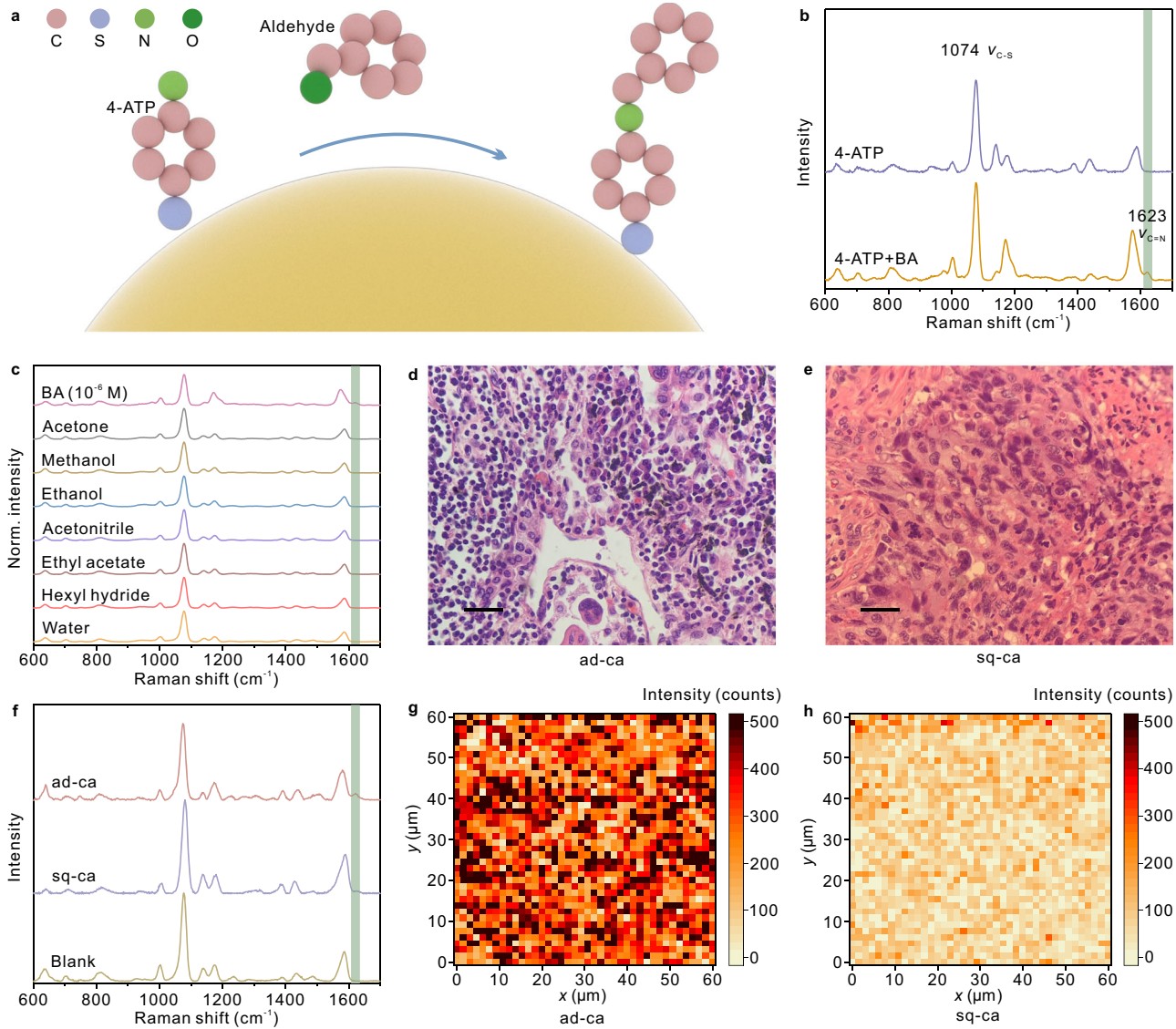

**Fig. 5 | Pathological diagnosis of lung tumors with the Au@Al₂O₃-Au-Au@Al₂O₃ trimer arrays. a** Schematic diagram of the reaction between aldehydes and 4-ATP on gold. **b** SERS spectra of 4-ATP on Au@Al₂O₃-Au-Au@Al₂O₃ trimers before (violet) and after (orange) reacting with discharged gas containing BA molecules. The C=N vibration mode suggesting the reaction is marked with a light green strip. **c** Anti-interference test of the Au@Al₂O₃-Au-Au@Al₂O₃ trimer samples exposed to BA, acetone, methanol, ethanol, acetonitrile, ethyl acetate, hexyl hydride and water. **d**, **e** Optical images of the ad-ca and sq-ca lung tumor tissues, respectively, after HE

staining under a microscope. The scale bars are 50 μm. This experiment was independently repeated for all the tumor samples, and the results were similar. **f** Typical SERS spectra from the samples containing fresh ad-ca (red) or sq-ca cancer (violet) tissues, and blank sample (brown). **g**, **h** Raman mapping at 1623 cm⁻¹ from the ad-ca and sq-ca samples (1600 points), respectively (step 1.5 μm, integral time 10 s), which color scales representing Raman intensities. Source data are provided as a Source Data file.

algorithm). The edge of the histogram indicates a high probability of single-molecule events. It can be observed that the contributions of $p \approx 1$ (4-Mpy event) and $p \approx 0$ (4-MBT event) are very close because of their similarity in Raman scattering cross sections. Moreover, single-molecule characteristics including signal blinking and spectrum wandering were investigated (see Supplementary Fig. 15), further identifying the validity of the single-molecule SERS events.

## Pathological diagnosis and subtyping of lung tumors

For lung cancer patients, the balances of oxygen free radical breakdown, leading to the lipid peroxidation of polyunsaturated fatty acids in cancer cell membranes and organelle membranes, and consequently extra production of aldehydes and other volatile organic compounds (VOCs)[55–57]. As a result, patients with lung cancer may

exhibit significantly higher levels of aldehydes in their lung tissues[58–60]. While SERS measurement of aldehyde molecules has been showcased with exhaled breath condensate from patients[61,62], conducting direct SERS analysis with human tissues remains challenging due to the limited quantity of aldehydes within cells. In-depth investigations, including methods for assessing the accuracy of SERS results with tissue biopsy data, exploring the relationship between TNM stage and SERS outcomes, and investigating the possibility of tumor subtyping by SERS, remains to be answered.

Leveraging the sensitivity of the Au@Al₂O₃-Au-Au@Al₂O₃ trimers, SERS investigations with human tissues were conducted to demonstrate their potential for the early diagnosis and subtyping of lung tumors, as depicted in Fig. 5. We prepared 4-ATP-modified Au@Al₂O₃-Au-Au@Al₂O₃ samples for the detection of aldehyde molecules. A

monolayer of 4-ATP was adsorbed on the gold trap particle via its thiol group during preparations. The amino group located at the opposite end of 4-ATP reacts with aldehydes to form a C=N bond for Raman identification[55,63], as illustrated in Fig. 5a. Typical Raman spectra from the 4-ATP-modified sample before and after benzaldehyde (BA) collection were displayed in Fig. 5b, where the mode at 1623 cm$^{-1}$ suggesting the C=N vibration mode can be clearly discerned after the collection[64,65]. The SERS optimizations for the experiment are available in Supplementary Fig. 16, with a detection limit of 10$^{-9}$ M for BA, as shown in Supplementary Fig. 17. Additionally, SERS detection of other aldehydes are presented in Supplementary Fig. 18. The anti-interference performance of the Au@Al$_2$O$_3$-Au-Au@Al$_2$O$_3$ trimer samples were displayed in Fig. 5c, suggesting the ability in specific detection of aldehyde molecules.

Further investigations were performed with fresh lung cancer tissue of different subtypes, as illustrated in Fig. 5d–h. Lung tumors are generally divided into adenocarcinoma (ad-ca), squamous carcinoma (sq-ca), and other minor subtypes[58]. We obtained ten fresh lung tumor tissues from patients by surgical operation. The tissues were cut into dimensions of approximately $1 \times 1 \times 0.5$ cm$^3$ for SERS investigations. The typical pathological characterizations of cancer cells from the two major subtypes after hematoxylin-eosin (HE) staining are shown in Fig. 5d, e. The 4-ATP modified trimer samples were immersed in buffer solution containing tumor tissues (Supplementary Fig. 19). We collected 1600 Raman spectra from each sample. The averaged SERS spectra from an ad-ca sample and a sq-ca sample are compared in Fig. 5f, suggesting that the signal intensity for ad-ca was much stronger than that for sq-ca. Corresponding Raman intensity maps are plotted in Fig. 5g, h, revealing that a majority of the measurement points was positive for ad-ca, while only a few events were positive for sq-ca.

We analyzed the SERS results from the ten patients, including six ad-ca, three sq-ca, and one benign case (Supplementary Fig. 20), and compared the results with their TNM stage (Supplementary Table 1). Some preliminary conclusions were drawn: 1) SERS is sensitive in distinguishing different lung tumor subtypes, suggesting variations in the body content of aldehyde molecules among them. Specifically, SERS is sensitive to ad-ca but not to sq-ca or benign cases. 2) There is no evidence suggesting a correlation between the SERS outcomes and the original tumor size or lymph node with the same volume of tumor tissue. For sq-ca patients, SERS diagnosis is appliable to all the patients with TNM stages from IA1 to IIB. Still, further studies with a larger dataset are required to refine these preliminary observations. Similar Raman measurement of ad-ca has also been also performed on the Au-Au-Au trimers for comparison, as revealed in Supplementary Fig. 21.

## Discussion

In summary, this study presents an approach to fabricate large-area nanoarrays composed of oriented plasmonic trimers. These trimers comprise a pair of plasmonic dimers with a trap particle positioned in between, selectively guiding probe molecules to the trap through chemical affinity. This configuration facilitates spatial overlap between the molecular adsorption sites and plasmonic 'hotspots' and enhances the probability that probe molecules experiencing the amplification from 'hotspots', offering a robust approach to improve SERS sensitivity against the heterogeneous 'hotspots'. We prepared the Au@Al$_2$O$_3$-Au-Au@Al$_2$O$_3$, Au-Au-Au, and other different types of trimers. The material of the plasmonic trimers can be modulated during the preparations, allowing on-demand construction of trimers with different compositions.

Benefiting from the trapping effects and small gap distances of the trimers, SERS performance was achieved. The sensitivity was demonstrated by comparing the SERS performance of the Au@Al$_2$O$_3$-Au-Au@Al$_2$O$_3$ and Au-Au-Au trimers using different molecules, including thiols, amines, disulfides and selenols, as evidenced by DFT calculations. Moreover, single-molecule sensitivity was demonstrated

with the bi-analyte method, further highlighting the SERS performance of the trimers.

The trimers were further applied in the pathological diagnosis of lung tumors. Leveraging the sensitivity of the trimers, SERS investigations were conducted with fresh lung tumor tissues, where the concentration of probe molecules is low. Our findings indicate that SERS is sensitive to ad-ca tumors, but not sq-ca or benign cases tumors, demonstrating the potential of SERS for the early diagnosis and subtyping of lung tumors. By addressing the crucial challenge posed by the heterogeneous distribution of 'hotspots', our approach offers an approach to explore the full potential of SERS in various applications.

## Methods

### Materials and chemicals

Au (99.999%), Ag (99.999%), titanium oxide (99.99%) and aluminum foil (99.999%) were purchased from Zhong Nuo Advanced Material Technology Co., Ltd (Beijing, China). The ALD precursor of trimethylaluminum (99.999%) and water (>18.2 MΩ cm) were purchased from Jiangsu MNT Micro and Nanotech Co, Ltd. Polymethyl methacrylate (PMMA) was purchased from MicroChem (950PMMA A4). Phosphoric acid (85 wt.%), perchloric acid (≥99.7%), oxalic acid (99%), chromic acid (5 wt.%), copper(II) chloride dihydrate (≥99.7%), acetone (≥99.7%), absolute ethanol (≥99.7%), 4-methylbenzenethiol (4-Mpy, 90%), 4-mercaptopyridine (4-MBT, 96%), 4-aminothiophenol (4-ATP, 90%), adenine (≥99.5%), thiram (97%), benzeneselenol (BSe, 97%), toluene (≥99.7%), 4-ethylbenzaldehyde (99.0%), 2-furaldehyde (99.0%), salicylaldehyde (99.0%), and benzaldehyde (99.0%) were purchased from Aladdin Biochemical Technology. All the reagents were used as received without further purification. Milli-Q water (>18.2 MΩ cm) was utilized in all experiments.

### Preparations of the AAO membranes

Briefly, electrochemically polished aluminum foils were anodized in 0.3 M oxalic acid at 40 V and 0 °C to fabricate the porous AAO membrane by a two-step anodization method[44]. The thickness of the fabricated membranes was 300 nm, the diameter of the pores was about 30 nm, and the center-to-center spacing between two adjacent pores was 100 nm. The fabricated AAO membranes were immersed in 5 wt.% phosphoric acid at 30 °C for 26 min to broaden the pore diameter. The broadened samples were spin-coated with PMMA at 1200 rpm for 60 s, and baked at 120 °C for 10 min. The AAO membranes were peeled off from the aluminum foil in CuCl$_2$ solution (10 wt.%), and transferred onto phosphoric acid solution (5 wt.%) at 30 °C for 36 min to further adjust the membrane thickness. Afterwards, the membranes were immersed in acetone to remove PMMA, and then transferred onto SiO$_2$/Si wafers. The final thickness of the membrane was about 240 nm, and the pore diameter was 80 nm.

### Fabrication of the trimer arrays

First, the dimer particles were fabricated by angle-resolved shadow deposition using AAO membranes[66]. The e-beam deposition was conducted at $7 \times 10^{-4}$ Pa and 0.1 nm s$^{-1}$ (Beijing Technol Science Co., Ltd). The deposition thicknesses were 35 nm, and the deposition angles for the first and second depositions were 8.7° and 7.9°, respectively. The AAO pores were partially blocked during the evaporations. Adjustments were made for the second deposition due to changes in pore diameters after the initial evaporation. Subsequently, a 1.5 nm thick Al$_2$O$_3$ layer was grown on the Au dimers by ALD (Jiangsu MNT Micro and Nanotech Co, Ltd.) according to ref.[67]. The ALD process is optional depending on the requirement. The diameter of the AAO pores was further adjusted to approximately 10 nm by magnetron sputtering (Beijing Chuangshiweina Technology Co., Ltd.). Following this, the trap particle was vertically deposited on the sample by e-beam deposition with a thickness of 20 nm. Ultimately, the trimer arrays were obtained by removing the AAO membranes with adhesive tape.

## Characterizations

The SEM images were obtained by FEI Inspect F50. The TEM images and elemental mappings were obtained by Thermo Fisher Talos F200X. The particles were transferred onto copper grids substrate by PMMA before TEM characterizations. The XRD patterns were obtained by a Rigaku Smartlab (3). The UV-vis spectra were collected by a HITACHI U-3900 spectrometer. The Raman spectra were collected by XploRA PLUS (HORIBA) with a 785 nm laser (spot diameter ≈ 1 μm), 100× objective (power density 0.5 mW), and 600 grooves mm$^{-1}$ grating. The images of the cancer cells were captured under a 40× optical microscope (OLYMPUS).

## Raman preparations

For 4-MBT, 4-Mpy, 4-ATP, thiram and BSe, as well as the mixture solution of 4-MBT and 4-Mpy, the SERS substrates were immersed in a 10 mL ethanol solution for 4 h for molecular adsorption, rinsed with ethanol and dried with nitrogen gas. For the treatment of adenine, the ethanol solution was replaced by an aqueous solution with all other conditions remaining constant. In the case of toluene, the substrates were immersed in a 10 mL solution for 4 h and directly dried with nitrogen gas, as toluene would be easily washed away due to its weak bonding with the substrates. In VOCs detection, the samples were modified with $10^{-6}$ M 4-ATP, and then placed in a closed container containing fresh lung cancer tissue in phosphate buffer (1× pH 7.4) at 55 °C for 30 min (see Supplementary Fig. 16). Fresh lung tumors tissues were provided by the Affiliated Hospital of Qingdao University from surgical operations. The tissues were cut into $1 × 1 × 0.5$ cm$^3$ fragments and immersed in sterile Earle's balanced buffer solution before SERS measurement. This study was approved by the Ethics Committee of the Affiliated Hospital of Qingdao University. All participates had signed the consent form. All methods were carried out in accordance with relevant guidelines and regulations. Informed consent was obtained from all subjects and/or their legal guardians.

## FEM simulations

The finite-element method (FEM) simulations were carried by COMSOL Multiphysics software. The excitation light is a linearly polarized plane wave perpendicular to the *x-y* plane. The complex refractive index ($n + ik$) of gold and aluminum oxide were obtained from the literature[68,69]. Scattering boundary conditions were applied to define the boundary. Periodic boundary conditions were employed, utilizing a lattice with a center-to-center distance of 100 nm, in reference to the AAO membrane lattice.

The parameters for FEM simulations were derived from statistical results obtained by TEM characterizations. For the Au@Al$_2$O$_3$-Au-Au@Al$_2$O$_3$ trimers, the long and short axes of the Au@Al$_2$O$_3$ ellipsoid are approximately 43 nm and 34 nm, respectively. The long and short axes of the trap particle are about 18 and 16 nm, respectively, with a statistical error of approximately 1-2 nm. The thickness of the Al$_2$O$_3$ coating is 1.5 nm ± 0.1 nm, and the gap between two adjacent particles is approximately 2 nm. The particles were modeled as 3D ellipsoids, and the height of the ellipsoidal dimer particles and the trap particle was set to 35 and 20 nm, respectively, in accordance with the e-beam evaporation parameters. Similar parameters were applied for the Au-Au-Au trimers, with the exception of the absence of the Al$_2$O$_3$ coating.

## DFT simulations

The DFT calculations were performed using the Vienna ab-initio simulation package (VASP) with projector augmented wave (PAW) method. The Perdew–Burke–Ernzerhof (PBE) exchange-correlation energy function was applied within the generalized gradient approximation (GGA). A cut-off energy of 500 eV for the plane-wave basis restriction was employed in all calculations, with a convergence criterion for the electronic self-consistent iteration set to be $10^{-5}$ eV. Gamma-only point sampling was used for Brillouin zone integration

during structure optimization. The geometry was fully relaxed without constraints until the force on each atom was less than 0.05 eV Å$^{-1}$. Grimme's DFT-D3 correction was incorporated to account for dispersion interactions. The slab model of Au(111) was generated based on the optimized crystal structure, with a vacuum layer of 15 Å introduced to ensure separation between slabs. The adsorption energy ($E_{ads}$) of an adsorbate A was defined as Eq. (1):

$$E_{ads} = E_{A+surf} - E_{surf} - E_A \qquad (1)$$

where $E_{A+surf}$, $E_{surf}$, and $E_A$ are the total energy of surface system, the energy of the surface slab, and the energy of adsorbate, respectively.

During the ALD process, amorphous and partially hydroxylated Al$_2$O$_3$ layers are deposited. In this study, γ-Al$_2$O$_3$ was employed for simulation due to its similarity to the alumina produced in the ALD process, exhibiting a poor crystalline nature[49,50]. Additionally, γ-Al$_2$O$_3$ shares similarities with amorphous alumina in having both 4-coordinated and 6-coordinated aluminum.

The crystallographic directions of γ-Al$_2$O$_3$ are typically given within the spinel orientation. Therefore, its thermodynamically most stable (100) surface within the spinel orientation was created from the γ-Al$_2$O$_3$ unit cell by adding a vacuum gap in the *z*-direction[51]. For the current calculations, an 8.4 Å × 16.1 Å × 20.8 Å box containing partially hydroxylated aluminum oxide layers was utilized[70]. The bottom 16 Al and 24 O atoms were kept fixed, while the top atoms of the slab and all the atoms of the adsorbed species were allowed to relax. This configuration includes terminal hydroxyl, bridge hydroxyl, and bridge oxygen on the surface. The adsorption modes of 4-MBT, 4-Mpy, 4-ATP, and toluene on 4- and 6-coordinated aluminum were considered, and the final results represent the most stable adsorption mode after optimization.

## Calculation of SERS enhancement factors

The experimental SERS enhancement factor was calculated according to Eq. (2)[1]:

$$EF = \left(\frac{I_{SERS}}{c_{SERS}}\right) / \left(\frac{I_{RS}}{c_{RS}}\right) \qquad (2)$$

Where $I_{SERS}$ and $I_{RS}$ are the SERS spectral intensity and intrinsic Raman spectral intensity, respectively, and $c_{SERS}$ and $c_{RS}$ are the corresponding molecular concentrations. Taking the Raman mode of 4-Mpy at 1098 cm$^{-1}$ as an example ($10^{-14}$ M, data extracted from Fig. 3), the EF of the Au@Al$_2$O$_3$-Au-Au@Al$_2$O$_3$ trimers was calculated to be ≈ $10^{10}$.

## Computational steps for MPCA algorithm

The steps of MPCA follow a standard method[71,72]: (1) 1640 Raman spectra were recorded, and 43 data points at the range of 1050−1120 cm$^{-1}$ of each spectrum were selected to define a matrix *M* of $T × N$ (1640 × 43); (2) Subtract the mean intensity from each row of matrix *M* to obtain matrix *M'*; (3) The covariance matrix *V* ($N × N$) of the matrix *M'*, and the eigenvalues and eigenvectors of the matrix *V* were calculated (Supplementary Fig. 14a). The first two eigenvectors in the statistical row of the dual analysis method are the dominating components; (4) A coefficient matrix *C* was obtained by operation, which is equivalent to the scalar product of the spectra and the first two eigenvectors. The matrix *C* was plotted in two-dimensional spatial coordinates (Supplementary Fig. 14b). Each point in the graph represents an event, with pure spectral events appearing on both the 4-MBT and 4-Mpy axes, null events near the origin, and mixed events appearing in the middle of the two axes; (5) Finally, linear transformation is employed to relocate the coordinates of the two axes in space. After removing the noise of the null event, the single-molecule

probability histogram $p$ was defined as Eq. (3):

$$p = \frac{1}{1 + \text{abs}(y/x)} \tag{3}$$

The $(x, y)$ are obtained through the original coordinates (Supplementary Fig. 14b). The histogram shows the contribution of different spectral events to the total number of spectra except for the null event.

## Reporting summary

Further information on research design is available in the Nature Portfolio Reporting Summary linked to this article.

## Data availability

The data that support the findings of this study are available from the corresponding author upon request. Source data are provided with this paper.

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

## Acknowledgements

Q.H. acknowledges the National Natural Science Foundation of China (Grant No. 22004016) and the Open Research Fund of Key Laboratory of Quantum Materials and Devices (Southeast University), Ministry of Education. T.Q. acknowledges the National Natural Science Foundation of China (Grant No. 12374370). X.F. acknowledges the Natural Science Foundation of Jiangsu Province (Grant No. BK20230807) and China Postdoctoral Science Foundation Funded Project (Grant No. 2021M700773). The authors acknowledge Southeast University and Key Laboratory of Quantum Materials and Devices of Ministry of Education for the technique support in simulations.

## Author contributions

Q.H. provided the original idea. Q.H. and T.Q. jointly supervised the project. X.Z. carried out the design of the experiment, prepared the measurements, and wrote the manuscript. X.L. and G.S. provided the lung tumor tissues and pathological analysis. X.F., X.T., M.L., L.Y., Y.W., W.S., Z.S. and Z.Z. participated in the characterizations and discussions. D.C., G.L. and X.N.Z. provided DFT and FEM simulation calculations, respectively. Q.H. corrected the manuscript. All authors have approved the final manuscript.

## Competing interests

The authors declare no competing interests.
