## [Peer Review File · Nature Communications]

Plasmonic trimers designed as SERS-active chemical traps for subtyping of lung tumorsReviewers' comments:

Reviewer #1 (Remarks to the Author):

Qi Hao et al. propose and fabricate heterogeneous plasmonic trimers consisting of a pair of silver dimers and a trapped particle in between to improve SERS sensitivity. They investigate the SERS performance of molecules containing thiols with the Ag-Au-Ag trimer arrays. The authors claim to reach single-molecule SERS sensitivity for molecules containing thiols, deduced from a by-analyte analysis. Furthermore, the authors demonstrate the potential for early detection of lung tumors by investigating the SERS signal of benzaldehyde (BA), including tumor tissues from surgery, showing different BA levels for different tumor subtypes. The experimental work is complemented by finite-element simulations (for the field enhancement prediction) and density functional theory calculations (for the binding energy between the molecules and the particle substrate). The work appears original and significant to the field. The experimental work seems to be comprehensive and methodologically sound. From a theoretical point of view, the authors provide minimal insight into their methods, which should be improved.

Consequently, I regard this work as publishable in Nature Communications only after major revisions. The authors should implement/comment on the following points:

A) The FEM method is not well described. The authors compare the near-field results to UV-VIS spectroscopic results. Still, they need a comprehensive review of the similarities and differences between the measurement and simulation setup. The authors need to address the following:

1) What software has been used?

2) What are the same material datasets used? "from the library" is insufficient. Specifically, the available Au datasets comprise a massive variance in literature, potentially leading to considerable mode shifts in the spectrum.

3) The simulation setup needs to be defined, and differences/similarities with the spectroscopic results should be critically discussed: How is the exact morphology of the simulated particle clusters (the experimental results indicate that the particles do not match perfect 3D-ellipsoid shapes)? What are the boundary conditions? What are the exact excitation conditions (except the polarization)? Do the simulations relate to single clusters or a periodic lattice? What about the inter-cluster energetic exchange? Etc.

4) The authors provide ellipsoid parameters and particle distances for their demonstrated FEM results. It needs to be made clear how these parameters arose. The particle morphology and the inter-particle distances play a crucial role in the resonance strength and position - have these parameters been optimized or just "guessed" from the microscopy images? More generally speaking, the manuscript would benefit significantly from a theoretical analysis of how the variance in morphological and material parameters influences the results (-> How stable is this approach against manufacturing tolerances?)

5) The simulation-deduced SERS enhancement factor seems to be extrapolated from the near-field results, but there is a way to explicitly calculate it [see, for instance, Charconnet et al. Small Methods]

B) The description of the underlying DFT is also insufficient, and a more comprehensive study is necessary to claim the final bonding energies. There are several open questions to be addressed:

6) Why are only Au and Ag 111 structures considered? The authors should provide evidence for this assumption since the metal interface with the analyte will significantly influence the bonding behavior.

7) How did the authors "reach" the shown "final" states for the bonding (and consequently the

bond(ing) energy)? How did the authors choose the initial conditions of the molecules near the surface? Were these positions swept or even optimized?

(As a side note, the document is well-readable, but it could benefit a lot from serious grammar checking before being published)

Reviewer #2 (Remarks to the Author):

In this manuscript, Hao and co-workers report a fabrication of Ag-Au-Ag nanotrimers and their SERS sensitivity. The authors prepared Ag-Au-Ag trimer arrays using an angle-resolved shadow deposition method and achieved a single-molecule level detection limit (100 fM) for thiol molecules. The high sensitivity of the heterotrimer nanostructures was applied to detect benzaldehyde (BA) from lung cancer tissues.

It is an interesting piece of work. I acknowledge that the authors made a tremendous amount of efforts to characterize the SERS properties of their nanotrimer system and find an application. They used various analytical (e.g., Modified PCA) and calculation methods (e.g., FEM and DFT) to corroborate their claim for the unprecedented SERS sensitivity of the heterotrimers. They tried to demonstrate the usefulness of their system by detecting BA that is a product of lung cancer activity. However, it still seems to me to be a mere extension of many nanoparticle assembly systems that usually have high SERS enhancements, compared to monomers, because of the presence of nanogaps (or "hot spots"). I don't see any evidence that this particular nanotrimer systems have larger SERS enhancement than other forms of nanoassemblies such as dimers, oligomers, or core-satellites, or other combinations of nanoassemblies such as cube-sphere, rod-sphere, star-sphere, or cube-cube. Furthermore, the authors' claim for the enrichment of analytes in nanogaps needs to be reevaluated. I don't think that a slight difference in binding energy of thiols to Au and Ag allows for a preferential adsorption of the molecule onto Au surfaces (see details in the comments below). Considering the nanoassembly system itself, the production method, SERS property, and even an application (see details in the comments below), I don't think this work is innovative enough to warrant its publication in Nature Communications.

Specific comments are as follows:

1. The authors fabricated Ag-Au-Ag trimer nanoarrays. Structural characterizations are necessary. The authors should provide the distribution of sizes of Ag nanoparticles, Au nanoparticles, and their gap distances.
2. The authors claim that the difference in binding energy of 4-MBA between Au (1.116 eV) and Ag (0.917 eV) leads to the preferential adsorption of the molecules onto Au nanoparticles ("trap nanoparticles") and thereby locates the molecules in hot spots between Ag nanoparticles and trap nanoparticles. I don't think that it is likely that the small binding energy difference (only 4.8 kcal/mol) determines the adsorption site of molecules. Rather, the authors should consider the energy barrier in the adsorption potential. Furthermore, the authors suggest that "strong chemical adsorption forms when the binding affinity is above 1.0 eV, and therefore, the 4-MBA were more likely to be chemically adsorbed on the gold trap particle" (line 181-184). Then, what about 4-ATP? The 4-ATP molecules have binding energies for Au (1.293 eV) and Ag (1.048 eV) - both above 1.0 eV. Then, do they bind to both Au and Ag? No selective binding?
3. Even if thiol molecules preferentially bind to Au trap nanoparticles, it doesn't guarantee that the molecules go into the narrow nanogap. They just might adsorb onto the open surface of Au nanoparticles upon the first collision. Therefore, the authors' claim for the enriched sample locations in nanogaps and the consequent high SERS enhancement is significantly undermined.
4. It is great to show that BA can be detected using SERS. However, the problem is that it is only possible through the reaction with 4-ATP. It is not a direct detection. The authors are detecting the imine product from the reaction between BA and 4-ATP. Thus, the authors must quantify first the reaction yield under various conditions. Because the reaction yield depends on the reaction conditions,

the solution test results may not be transferrable to the cancer tissue cases.

5. Another feature of the SERS spectra arising from a single molecule is that vibrational frequency varies depending on the local environment and a conformation of the molecule. In addition to the bianalyte experimental results in Figure 5b, the vibrational spectral diffusion might make the authors' claim for the single-molecule SERS more convincing.

6. Minor typographical errors:

- Line 135: Abbreviations must be written in full upon first appearance.
- Line 153: horizontal excitation corresponds to $\theta = 0^\circ$, not 90° .
- Line 296: Ag-Au-Au -> Ag-Au-Ag

Reviewer #3 and #4, who co-reviewed the work (Remarks to the Author):

The authors present the nanofabrication of a heterotrimer comprising a central gold nanostructure surrounded by a larger silver nanostructure left and right next to it. The nanofabrication is based on elegant approach using a AAO mask in combination with different types of deposition. Overall, the nanostructure has a length of ca. 100 nm (SEM image in Fig. 3A). The key idea of using this plasmonic Ag-Au-Ag heterotrimer for improved SERS sensitivity is to exploit the different affinities of target analytes towards gold versus silver. Two SERS applications are presented: single-molecule/few molecules SERS and lung tumor diagnosis.

The main strength of this contribution is the nanofabrication part for obtaining such Ag-Au-Ag heterotrimers. However, with respect to SERS, there are some fundamental limitations inherent to this approach. First, as in most regular SERS applications, thiol-containing analytes are detected. This is quite boring. The current challenge in SERS is to detect other than these trivial analytes without surface-seeking groups. The presented enhancement of the structure is questionable considering the relatively large volume of the hot spot around the central gold nanostructure. The accompanying FEM simulations were performed on an object for which the dimensions are not specified, i.e., they might not correspond to the experimental situation. In the SERS application for lung tumor diagnosis 4-ATP-functionalized gold nanostructure for capturing aldehyds in order to form the corresponding Schiff bases/imines (C=N) are employed. Thus, in principle any aldehyde and not only benzaldehyde (BA) claimed in this study could be detected. In other words, there is no inherent selectivity in this approach.

Overall, several experimental and theoretical details are missing, which makes it hard to believe the presented claims.

Since this work is not addressing the broad and heterogenous readership of Nature Communications, but contains a really nice nanofabrication part (the SERS applications lack originality and novelty), we recommend sending this contribution to a journal devoted to nanoscience or material science.

Reviewer #5 and #6, who co-reviewed the work (Remarks to the Author):

In this manuscript, the authors investigate an approach to improve the sensitivity of surface-enhanced Raman scattering (SERS) for molecular sensing by using heterogeneous plasmonic trimers.

Conventionally, SERS suffers from the fact that only molecules in electromagnetic hot spots, which constitute only a small proportion of the total molecules, contribute to the SERS signal. The proposed trimers increase the concentration of the probe molecules in these hot spots through chemical affinity, resulting in a significant improvement in SERS sensitivity.

Furthermore, the authors demonstrate the potential of this technique in early detection of lung tumors by analyzing the SERS signal of benzaldehyde. The pathological diagnosis of different tumor subtypes is illustrated based on differences in SERS intensity, which indicates that the levels of benzaldehyde can be different in different tumor subtypes.

While the manuscript presents interesting results and is well-written with high-quality figures, the novelty of the findings does not warrant publication in Nature Communications.

There has already been extensive work investigating the concentration of target molecules in electromagnetic hotspots, such as in the following two references:

- Victoria M. Szlag, Rebeca S. Rodriguez, Jiayi He, et al. ACS Applied Materials & Interfaces 10 (38), 31825-31844 (2018).
- De Angelis, F., Gentile, F., Mearini, F. et al. Nature Photonics 5, 682–687 (2011).

Similarly the use of bowtie-like structures to generate strong hot spots is well explored, see for example the following reference:

- For photothermal applications: Pratiksha D. Dongare, Yage Zhao, David Renard, Jian Yang, Oara Neumann, Jordin Metz, Lin Yuan, Alessandro Alabastri, Peter Nordlander, and Naomi J. Halas ACS Nano 15 (5), 8761-8769 (2021).
- For spectroscopy: M. Kaniber, K. Schraml, A. Regler, J. Bartl, G. Glashagen, F. Flassig, J. Wierzbowski & J. J. Finley, Scientific Reports volume 6, Article number: 23203 (2016)
- For lasing: Jae Yong Suh, Chul Hoon Kim, Wei Zhou, Mark D. Huntington, Dick. Co, Michael R. Wasielewski, and Teri W. Odom, Nano Lett. 2012, 12, 11, 5769–5774

Even plasmonic trimers themselves have been previously reported for Raman scattering:

- For Raman scattering: Shuangmei Zhu, Chunzhen Fan³, Erjun Liang, Pei Ding, Xiguang Dong, Haoshan Hao, Hongwei Hou & Yuanda Wu, Scientific Reports volume 11, Article number: 1230 (2021)
- For nonlinear optics: Seyfollah Toroghi, Chatdanai Lumdee, and Pieter G. Kik, Appl. Phys. Lett. 106, 103102 (2015)

Furthermore, it is not clear how general this approach is. What if the target molecules do not have a high chemical affinity for gold? Replacing active functionalization with ligands with the intrinsic chemical affinity of the particles is not a sufficiently novel or general approach. Also the impact on lung tumor diagnosis is not clear: how does that compare with other techniques? In line 274, the authors state: "it was found that the identification of lung cancer with human breath is not always successful even for the patients which have been confirmed with lung tumors". How the presented technique advances the state of the art for detection?

For these reasons, Nature Communications is not the proper venue for this manuscript.

From a technical point of view, the authors could also provide further details in certain places. For example, in lines 191-195, the authors state that the observed SERS enhancement of the Ag-Au-Ag trimer should not arise from the coupling between plasmonic modes and the laser wavelength. In order to show this, the field enhancement between the Ag-Au-Ag and Ag-Ag-Ag trimers should be

compared. However, the authors only show the field enhancement Ag-Au-Ag trimers in Figure 3b, while in Supplementary Figure 4, they show only the extinction spectra and not the field enhancement.

Point-by-Point Replies to the referees: (turn #1, NCOMMS-23-09102A-Z, 10th December 23, 2023)

Overall Modifications :

I would like to express our sincere gratitude to the reviewers for the constructive feedback and insightful comments provided during the review process of our manuscript. We have meticulously addressed each comment in this response letter and implemented necessary modifications in the revised manuscript.

Specifically, we have replaced the original Ag-Au-Ag trimer configuration with the Au@Al₂O₃-Au-Au@Al₂O₃ trimers. While the logical framework of the paper remains unchanged, this modification necessitated a comprehensive overhaul of various aspects, including sample characterizations, SERS measurements, FEM and DFT simulations, and biological applications. The introduction of the Au@Al₂O₃-Au-Au@Al₂O₃ trimer configuration addresses the concerns raised by the reviewer in the following ways: 1) Preferential Adsorption: The substantial differences between alumina and gold result in the preferential adsorption of thiol molecules on the gold trap particle. 2) Efficient Separation: The alumina ALD shell efficiently separates adjacent metallic particles, enabling the fabrication of trimers with smaller gap distances (~ 2 nm) without concerns about the connection between adjacent particles. Following modifications have also been included to address this issue and the concerns from the reviewer:

- 1) Precise TEM characterizations and comprehensive DFT simulations to support the viewpoint of selective adsorption (Fig. 2, Fig. S3, S7 and S8).
- 2) Updated SEM images, Raman probes, SERS measurement and single-molecule studies to align with and substantiate the changes in trimer configurations. (Fig. 3, Fig 4, Fig. S4, S6, S9, S10, S11 and S14)
- 3) Fabrication of different types of trimers to illustrate the generality (Fig. S3).
- 4) Precise structural characterizations (from TEM) for FEM simulations. (Fig. 2, Fig S5, Methods Section)
- 5) Blinking evidence for the identification of single-molecule events. (Fig. S12)
- 6) Quantification of the reaction yield (Fig. S13)
- 7) SERS spectra of different aldehyde molecules for a more comprehensive conclusion. (Fig. S15)
- 8) Revised and expanded investigations of the subtyping of lung tumors with more data (1600 spectra from each of the 10 tumor samples). (Fig 5, Fig S17)

- 9) Additional experimental results, and discussions on the relationship between tumor node metastasis (TNM) stage and SERS outcomes. (Fig 5, Table S1)
- 10) Supplemented discussions regarding the significance of this paper: 1) Clarifications about the advantages of our fabrication strategy; 2) Clarification about the significance of our presented techniques compared with the state of the art on lung tumor diagnosis.

It is noteworthy that the entire paper has been rewritten, except for the introduction and abstract, due to the replacement of the trimers. **Considering the tremendous modifications made to the manuscript, only the key sections were highlighted in red.**

Reviewer: 1

Overall comment: Qi Hao et al. propose and fabricate heterogeneous plasmonic trimers consisting of a pair of silver dimers and a trapped particle in between to improve SERS sensitivity. They investigate the SERS performance of molecules containing thiols with the Ag-Au-Ag trimer arrays. The authors claim to reach single-molecule SERS sensitivity for molecules containing thiols, deduced from a by-analyte analysis. Furthermore, the authors demonstrate the potential for early detection of lung tumors by investigating the SERS signal of benzaldehyde (BA), including tumor tissues from surgery, showing different BA levels for different tumor subtypes. The experimental work is complemented by finite-element simulations (for the field enhancement prediction) and density functional theory calculations (for the binding energy between the molecules and the particle substrate). The work appears original and significant to the field. The experimental work seems to be comprehensive and methodologically sound. From a theoretical point of view, the authors provide minimal insight into their methods, which should be improved. Consequently, I regard this work as publishable in Nature Communications only after major revisions.

Reply: We would like to thank the reviewer for thorough review of the manuscript and constructive comments. We have diligently addressed the raised points and made the necessary revisions to align the manuscript with the suggestions, specifically incorporating references from Charconnet et al. Below, we provide a detailed account of our responses and the corresponding modifications made to the manuscript.

Q1.1: What software has been used?

Reply: COMSOL Multiphysics software was used for all the FEM simulations. Relative descriptions have been added in the revised manuscript (Methods section, Page 17).

Q1.2: What are the same material datasets used? “from the library” is insufficient. Specifically, the available Au datasets comprise a massive variance in literature, potentially leading to considerable mode shifts in the spectrum.

Reply: The complex refractive index ($n + ik$) of gold and alumina were obtained from the literature [Appl. Opt. 37, 5271 (1998); Phys. Rev. Lett. 115, 177402 (2015)]. These parameters were adopted because it matches well with the experimental results and FDTD simulations of our particle arrays [J. Phys. Chem. Lett. 13, 7816 (2022)].

Relative descriptions have been added in the revised manuscript (Page 17).

Q1.3: The simulation setup needs to be defined, and differences/similarities with the spectroscopic results should be critically discussed: How is the exact morphology of the simulated particle clusters (the experimental results indicate that the particles do not match perfect 3D-ellipsoid shapes)? What are the boundary conditions? What are the exact excitation conditions (except the polarization)? Do the simulations relate to single clusters or a periodic lattice? What about the inter-cluster energetic exchange? Etc.

Reply: We have reperfomed the simulations, setting the parameters based on the statistical results from TEM characterizations (see **Q1.4**). For FEM simulations, the particles were approximated as 3D semi-ellipsoids. The height of the ellipsoidal dimer particles and trap particle were set to be 35 nm and 20 nm, respectively, based on the E-beam evaporation parameters. Plane wave excitations were applied with a wave vector perpendicular to the X-Y plane. Scattering boundary conditions were applied to define the boundary. Periodic boundary conditions were employed, utilizing a lattice with a center-to-center distance of 100 nm, in reference to the AAO membrane lattice (Nanotechnology, 2017, 28(10): 105301). Relative descriptions have been added in the revised manuscript (Page 17).

Q1.4: The authors provide ellipsoid parameters and particle distances for their demonstrated FEM results. It needs to be made clear how these parameters arose. The particle morphology and the inter-particle distances play a crucial role in the resonance strength and position - have these parameters been optimized or just “guessed” from the microscopy images? More generally speaking, the manuscript would benefit significantly from a theoretical analysis of how the variance in morphological and material parameters influences the results (-> How stable is this approach against manufacturing tolerances?)

Reply: Thanks for your valuable comments. In the revised manuscript, the FEM simulation parameters were set based on the following experimental parameters: According to the TEM

statistical results, the long and short axes of the Au@Al₂O₃ ellipsoid are 43 nm and 34 nm, respectively, with a statistical error of <2 nm. The long and short axes of the trap particle are about 18 and 16 nm, respectively, with a statistical error of about 1 nm. The thickness of the Al₂O₃ ALD coating is 1.5 nm ± 0.1 nm. The gap between two adjacent particles is approximately 2 nm. Similar experimental parameters were applied for the Au-Au-Au trimers, except for the absence of the Al₂O₃ coating.

The lattice of the AAO pores is 100 nm, which is rigorously determined by the anodization voltage. The thickness of the AAO pores can be well controlled with a statistical error of ~10 nm (Nanotechnology, 2017, 28(10): 105301). While the AAO mask exhibits nice stability and reproducibility, we acknowledge its lack of long-range order due to the non-single-crystal nature of the aluminum foil used for anodization, as depicted in the large-area SEM image (Figure S4). This may induce extra LSPRs coupling between different trimer units, a factor challenging to account for in simulations. Regarding the manufacturing tolerances, it may lead to variations in the peak of LSPRs modes within ~5 nm for different samples based on our experience. Besides, considering that the size of the laser spot is approximately 1 μm, while the scale of the trimer unit is 100 nm, the variations in LSPRs between different units would be significantly averaged.

We remain open to further modifications if additional comments arise during the review process. Relevant information has been added into the revised manuscript (Page 17).

Q1.5: The simulation-deduced SERS enhancement factor seems to be extrapolated from the near-field results, but there is a way to explicitly calculate it [see, for instance, Charconnet et al. Small Methods]

Reply: Thanks for the suggestion. We have implemented the necessary revisions in the manuscript. The relevant paper has been included in the references for further clarity and context. (Page 8)

Q1.6: Why are only Au and Ag 111 structures considered? The authors should provide evidence for this assumption since the metal interface with the analyte will significantly influence the bonding behavior.

Reply: In the revised manuscript, we investigated Au(111) and γ-Al₂O₃(100) instead. The XRD and HR-TEM results indicate that the exposed facets of the gold nanoparticles preferred to be Au (111), as depicted in **Figure R1**:

Fig. R1 (revised Supplementary Fig. 7). **a** XRD patterns of the Au@Al₂O₃-Au-Au@Al₂O₃ sample on silicon wafer substrate. The XRD patterns reveal a distinct peak at 38.2°, attributed to the (111) facets of fcc Au (JCPDS No. 04-0784). **b** HR-TEM image of a gold nanoparticle produced by E-beam evaporation. The interplanar spacing of 0.235 nm was assigned to the (111) facet of face-centered (fcc) Au.

As for γ -Al₂O₃(100), the reasons are explained as following: Amorphous and partially hydroxylated Al₂O₃ layers exhibiting a poor crystalline nature are deposited during ALD. In this study, γ -Al₂O₃ was employed for simulation due to its similarity to the alumina produced in the ALD process [J. Phys. Chem. C 123, 485 (2019); Nat. Commun. 10, 3139 (2019)]. Additionally, γ -Al₂O₃ shares similarities with amorphous alumina in having both 4-coordinated and 6-coordinated aluminum. The crystallographic directions of γ -Al₂O₃ are typically given within the spinel orientation. Therefore, its thermodynamically most stable (100) surface within the spinel orientation was created from the γ -Al₂O₃ unit cell by adding a vacuum gap in the z-direction [J. Phys. Chem. C 119, 13050 (2015)]. Considering that the alumina would be partially hydroxylated during the experiments [Science 288, 1029 (2000)], an 8.4 Å × 16.1 Å × 20.8 Å box containing partially hydroxylated aluminum oxide layers was ultimately utilized for the simulations. **Relative information has been added in the revised manuscript. (Methods section, Page 18)**

Q1.7: How did the authors “reach” the shown “final” states for the bonding (and consequently the bond(ing) energy)? How did the authors choose the initial conditions of the molecules near the surface? Were these positions swept or even optimized?

Reply: Thanks for your comments. In response to your query on the adsorption configurations of 4-Mpy on Au(111), we initially explored three distinct adsorption sites: top, bridge, and hollow. These initial configurations were subsequently subjected to geometry optimization in order to identify the most stable adsorption configuration. The optimizations primarily involved the adjustment of atomic coordinates to minimize the total energy of the system, a fundamental step in DFT calculations. To model the Au(111) surface, we employed a 4-layers (5×5) cell configuration. The lower two layers were confined to the simulated main region, while the upper two layers and adsorbents were allowed to relax, effectively representing the interface region. To prevent

unexpected interactions between periodic images during the global optimization process, we introduced a vacuum slab with a thickness of 15 Å in the Z direction. The simulation results are shown in the **Figure R2** (revised Supplementary Fig. 8):

Fig. R2 (revised Supplementary Fig. 8) DFT calculation results for 4-Mpy, 4-MBT, 4-ATP and toluene. a Adsorption energy at different adsorption sites for 4-Mpy (a), 4-MBT (b), 4-ATP (c), and toluene (d), respectively, on Au(111) and γ -Al₂O₃(100).

Q1.8. As a side note, the document is well-readable, but it could benefit a lot from serious grammar checking before being published.

Reply: We apologize for the grammatical errors in the document. We have conducted a careful review before this new submission.

Reviewer: 2

Overall comments: In this manuscript, Hao and co-workers report a fabrication of Ag-Au-Ag nanotrimers and their SERS sensitivity. The authors prepared Ag-Au-Ag trimer arrays using an angle-resolved shadow deposition method and achieved a single-molecule level detection limit (100 fM) for thiol molecules. The high sensitivity of the heterotrimer nanostructures was applied to detect benzaldehyde (BA) from lung cancer tissues.

It is an interesting piece of work. I acknowledge that the authors made a tremendous amount of efforts to characterize the SERS properties of their nanotrimer system and find an application. They used various analytical (e.g., Modified PCA) and calculation methods (e.g., FEM and DFT) to corroborate their claim for the unprecedented SERS sensitivity of the heterotrimers. They tried to demonstrate the usefulness of their system by detecting BA that is a product of lung cancer activity. However, it still seems to me to be a mere extension of many nanoparticle assembly systems that usually have high SERS enhancements, compared to monomers, because of the presence of nanogaps (or “hot spots”). I don’t see any evidence that this particular nanotrimer systems have larger SERS enhancement than other forms of nanoassemblies such as dimers, oligomers, or core-satellites, or other combinations of nanoassemblies such as cube-sphere, rod-sphere, star-sphere, or cube-cube. Furthermore, the authors’ claim for the enrichment of analytes in nanogaps needs to be reevaluated. I don’t think that a slight difference in binding energy of thiols to Au and Ag allows for a preferential adsorption of the molecule onto Au surfaces (see details in the comments below). Considering the nanoassembly system itself, the production method, SERS property, and even an application (see details in the comments below), I don’t think this work is innovative enough to warrant its publication in Nature Communications.

Reply to Overall Comments: We appreciate the thorough review of our manuscript and value the constructive comments provided by the reviewer. We acknowledge the major concerns raised, particularly regarding the ability to enrich molecules based on the differences in bonding affinities in the trimer configuration and the perceived lack of evidence demonstrating the SERS sensitivity. We find these concerns to be valid and have undertaken substantial revisions in response to your feedback.

In the revised manuscript, **we have replaced the original Ag-Au-Ag trimer configuration with the Au@Al₂O₃-Au-Au@Al₂O₃ trimers.** While the logical framework of the paper remains unchanged, this modification necessitated a comprehensive overhaul of various aspects, including sample characterizations, SERS measurements, FEM and DFT simulations, and biological applications. The introduction of the Au@Al₂O₃-Au-Au@Al₂O₃ trimer configuration addresses the concerns raised by the reviewer in the following ways: **1) Preferential Adsorption:** The substantial

differences between alumina and gold result in the preferential adsorption of thiol molecules on the gold trap particle. **2) Efficient Separation:** The alumina ALD shell efficiently separates adjacent metallic particles, enabling the fabrication of trimers with smaller gap distances (~ 2 nm) without concerns about the connection between adjacent particles. The characterizations of the Au@Al₂O₃-Au-Au@Al₂O₃ trimers is shown in **Figure R3** (revised Fig. 2):

Fig. R3 (revised Figure 2). Characterizations of the Au@Al₂O₃-Au-Au@Al₂O₃ trimers.

To substantiate these changes, **1)** we have included precise TEM characterizations of the samples and comprehensive DFT simulations (Supplementary Fig. 8) in the revised manuscript, with the aim to provide a clearer and more robust presentation of our findings. **2)** Moreover, different trimers, including Au-Au-Au, Au-Ag-Au, Au@Al₂O₃-Au-Au@Al₂O₃, and Au-TiO_x-Au, were prepared to illustrate the generality of this approach (Supplementary Fig. 3). **3)** Additionally, we utilized a newly acquired Raman spectrometer, which provides higher sensitivity and the capability for rapid mapping for SERS measurements. It enables in-depth investigations of the subtyping of the lung tumors, and explorations of the relationship between tumor node metastasis (TNM) stage with SERS outcomes (revised Fig. 5).

Our approach holds several advantages, including 1) ultrasmall gaps, 2) large-area fabrication, 3) free from surface chemicals, 4) aligned orientations, and 5) adjustable components. These unique features distinguish our approach from comparable techniques like EBL and colloidal methods. We believe that these modifications significantly enhance the manuscript, and we appreciate the opportunity to improve the quality of our work based on the further feedback received from the reviewer.

Q2.1: The authors fabricated Ag-Au-Ag trimer nanoarrays. Structural characterizations are necessary. The authors should provide the distribution of sizes of Ag nanoparticles, Au nanoparticles, and their gap distances.

Reply: We have included precise TEM characterizations in the revised manuscript. According to the TEM analytical results, the long and short axes of the Au@Al₂O₃ ellipsoid are 43 nm and 34 nm, respectively, with a statistical error of <2 nm. The long and short axes of the trap particle are about 18 and 16 nm, respectively, with a statistical error of about 1 nm. The thickness of the Al₂O₃ ALD coating is 1.5 nm ± 0.1 nm. The gap between two adjacent particles is approximately 2 nm.

Q2.2: The authors claim that the difference in binding energy of 4-MBA between Au (1.116 eV) and Ag (0.917 eV) leads to the preferential adsorption of the molecules onto Au nanoparticles (“trap nanoparticles”) and thereby locates the molecules in hot spots between Ag nanoparticles and trap nanoparticles. I don’t think that it is likely that the small binding energy difference (only 4.8 kcal/mol) determines the adsorption site of molecules. Rather, the authors should consider the energy barrier in the adsorption potential. Furthermore, the authors suggest that “strong chemical adsorption forms when the binding affinity is above 1.0 eV, and therefore, the 4-MBA were more likely to be chemically adsorbed on the gold trap particle” (line 181-184). Then, what about 4-ATP? The 4-ATP molecules have binding energies for Au (1.293 eV) and Ag (1.048 eV) - both above 1.0 eV. Then, do they bind to both Au and Ag? No selective binding?

Reply: We thank the reviewer for the comments and insights. While a previous study has indicated the preferential adsorption of thiols on gold rather than silver [J. Phys. Chem. Lett. 11, 1022 (2020)], we acknowledge the necessity for a more convincing presentation of our claims. To address this concern, the original trimer configuration is replaced by Au@Al₂O₃-Au-Au@Al₂O₃ trimers in the revised manuscript, as briefly outlined in the **reply to overall comments**. Here we aim to clarify that the molecules containing thiols preferentially adsorb onto the Au trap particles in the Au@Al₂O₃-Au-Au@Al₂O₃ trimers. The details are explained as follows:

1) Preliminary proof: Generally, thiol adsorbs to metal but not oxides. This is firstly examined by our preliminary experiments by comparing the SERS performance between the monomer Au NPs arrays and monomer Au@Al₂O₃ NPs arrays. **Figure R4** illustrates that 4-Mpy, a molecule with a thiol group, shows better performance on the AuNPs, while no signal is observed on the Au@Al₂O₃ NPs. In contrast, benzoic acid, a molecule with a carboxy group, exhibits better SERS performance on the Au@Al₂O₃ NPs. The results indicate that thiol molecules do not adsorb on the Au@Al₂O₃ due to the presence of the alumina shell.

Fig. R4. SERS spectra of 4-Mpy and benzoic acid, respectively, on the monomer Au NPs arrays and Au@Al₂O₃ NPs arrays. (The monomer nanoparticle arrays were fabricated simply by a direct vertical E-beam evaporation with the AAO mask.)

2) DFT simulations: We employed DFT simulations for a rigorous examination of our viewpoint. The adsorption energy of different molecules, including 4-Mercaptopyridine (4-Mpy), 4-Methylbenzenethiol (4-MBT) and 4-Aminothiophenol (4-ATP) and toluene on Au(111) and γ -Al₂O₃(100), were calculated (Supplementary Fig. 8). The expose facets of gold were determined by XRD and HRTEM characterizations (Supplementary Fig. 7), while the model for Al₂O₃ was established according to the results reported in the literature (Methods section). The results support our viewpoint that molecules with thiols would be preferably adsorbed on gold in the Au@Al₂O₃-Au-Au@Al₂O₃ trimers. (further details have been addressed in our response to Reviewer #1).

3) SERS performance. The performance of the Au@Al₂O₃-Au-Au@Al₂O₃ trimers was compared with Au-Au-Au, employing molecules with a thiol group (4-Mpy) and without it (toluene). The results reveal a significant improved SERS performance for 4-Mpy on the Au@Al₂O₃-Au-Au@Al₂O₃ trimers (Fig. R5a). Meanwhile, no distinct differences in SERS intensities were observed for toluene between the two cases (Fig. R5b), as toluene adsorbs on gold and Al₂O₃ with similar binding affinities due to a methyl group.

Fig. R5 SERS spectra of 10^{-9} M 4-Mpy (a) and toluene (b), respectively, from the Au@Al₂O₃-Au-Au@Al₂O₃ and Au-Au-Au trimers (extracted from revised Figure 3).

Q2.3: Even if thiol molecules preferentially bind to Au trap nanoparticles, it doesn't guarantee that the molecules go into the narrow nanogap. They just might adsorb onto the open surface of Au nanoparticles upon the first collision. Therefore, the authors' claim for the enriched sample locations in nanogaps and the consequent high SERS enhancement is significantly undermined.

Reply: We would like to address this question from three perspectives as follows:

First, though thiol molecules were conventionally assumed to be adsorbed on any surface site on the nanoparticles in principle, recent reports suggest that at low concentrations, they preferentially adsorb on the high-curvature surfaces of gold nanoparticles [Langmuir 39, 15828 (2023)], as illustrated in **Figure R6**.

Fig. R6 Langmuir adsorption isotherms of 4-MBA on gold nanoparticles. Reproduced with permission from {Langmuir 39, 15828 (2023)}. Copyright {(2023)} {American Chemical Society}.

Second, even if molecules uniformly adsorb on the surface, the trimer configuration can enhance the probability of a molecule being located in a hotspot. The distributions of the electromagnetic fields around the trimers are depicted in **Figure R7**. Assuming that sites with E/E_0 higher than 20 are considered hotspots, the probability of a site being a hotspot is 18.1% for the Au@Al₂O₃-Au-Au@Al₂O₃ trimers ($S_{\text{traphot}}/S_{\text{trap}}$), whereas this probability for Au-Au-Au trimers is only 1.1% ($S_{\text{totalhot}}/S_{\text{total}}$). This difference primarily arises from variations in the amount of surface area between the particles. Though there is no absolute standard in determining the quantity of hotspots, the comparison provides an indication that the probability of an adsorbed molecule experiencing significantly higher enhancement is greater on Au@Al₂O₃-Au-Au@Al₂O₃ trimers.

Fig. R7 3D waterfall map representing the EM intensity around the trimers.

Third, trap particles with a smaller size were employed in the revised manuscript, addressing the potential concerns from by the reviewer. The diameter of the trap particle was decreased by approximately 5 nm, further enhancing the proportion of hotspot area on the trap particle.

Overall, we have provided some evidence to support our viewpoint. As a matter of fact, we admit that the conditions raised by the reviewer cannot be completely ruled out despite existing literature that supports our perspectives. Nevertheless, the preferential adsorption of thiols on gold remains intact. We hope our response is received in a respectful manner.

(In fact, we are aware of the question raised by the reviewer, which has been illustrated by a nice paper by Eric C. Le Ru et al. [J. Am. Chem. Soc. 136, 10965–10973 (2014)]. However, we are not certain that if this issue exactly exists in arrays substrates, as there seems to be a conflict to our raised reference [Langmuir 39, 15828 (2023)]. If this concern indeed exists, it appears to be a general challenge applicable to a broad range of array substrates rather than exclusive to ours. At the present state, we have carried all possible efforts to handle this matter.)

Q2.4: It is great to show that BA can be detected using SERS. However, the problem is that it is only possible through the reaction with 4-ATP. It is not a direct detection. The authors are detecting the imine product from the reaction between BA and 4-ATP. Thus, the authors must quantify first the reaction yield under various conditions. Because the reaction yield depends on the reaction conditions, the solution test results may not be transferrable to the cancer tissue cases.

Reply: We thank the reviewer for the comments. The reaction yields with different reaction time and different temperatures were conducted in response to your feedback, as depicted in **Figure R8** (Supplementary Fig.13).

Fig. R8 Raman spectra of 10^{-9} M BA molecules with a different reaction time at 55 °C and with a different temperature for 30 min (Supplementary Fig. 13).

We are sorry that similar experiments cannot be carried out with the tissues. The timing constraint arises from the necessity to perform SERS measurements immediately upon receiving the sample,

even before obtaining pathological results from the hospital. Consequently, we cannot definitively determine whether the sample is an ad-ca or sq-ca sample before the SERS measurement, which poses challenges for conducting reaction yield experiments. Given the precious and limited nature of tumor samples, we opted for this general chemical reaction. We hope the reviewer can appreciate the constraints and limitations of our experimental setup. Relative discussions have been added in the revised manuscript and supporting information file (Supplementary Fig. 13).

Q2.5: Another feature of the SERS spectra arising from a single molecule is that vibrational frequency varies depending on the local environment and a conformation of the molecule. In addition to the bianalyte experimental results in Figure 5b, the vibrational spectral diffusion might make the authors' claim for the single-molecule SERS more convincing.

Reply: We have revised our manuscript according to the suggestions. The time-dependent Raman map of a 4-MBT single-molecule SERS event was shown in **Figure R9**. The single-molecule event was sort out by the bi-analyte method. The results indicate a clear blinking phenomenon, signifying a high probability of a single-molecule event. However, the spectra wandering was not very clear. We surmise that this may arise from the adsorption orientation of the molecule and the strong bonding between the molecule and substrate. Relative discussions have been incorporated into the revised manuscript and supporting information file (Supplementary Fig. 12).

Fig. R9 Time-dependent SERS map of a 4-MBT single molecule event (Supplementary Fig. 12).

Q2.6: Minor typographical errors:

- Line 135: Abbreviations must be written in full upon first appearance.
- Line 153: horizontal excitation corresponds to $\theta = 0^\circ$, not 90° .
- Line 296: Ag-Au-Au -> Ag-Au-Ag

Reply: We have revised the manuscript accordingly.

Reviewer: 3 & 4

Overall Comment: The authors present the nanofabrication of a heterotrimer comprising a central gold nanostructure surrounded by a larger silver nanostructure left and right next to it. The nanofabrication is based on elegant approach using a AAO mask in combination with different types of deposition. Overall, the nanostructure has a length of ca. 100 nm (SEM image in Fig. 3A). The key idea of using this plasmonic Ag-Au-Ag heterotrimer for improved SERS sensitivity is to exploit the different affinities of target analytes towards gold versus silver. Two SERS applications are presented: single-molecule/few molecules SERS and lung tumor diagnosis.

Reply to Overall Comment: Before addressing the specific questions, we would like to underscore the substantial modifications made to the manuscript: **We have replaced the original Ag-Au-Ag trimer configuration with the Au@Al₂O₃-Au-Au@Al₂O₃ trimers**, with a reduced gap distance to ~2 nm. This alteration, driven by the substantial differences between alumina and gold, ensures the preferential adsorption of thiol molecules on the gold trap particle. While the logical framework of the paper remains unchanged, this modification necessitated a comprehensive overhaul of various aspects:

1) **Sample Characterizations:**

HAADF-STEM, TEM: High-resolution characterization of morphology,

HRTEM and XRD: Revealing exposed facets of gold nanoparticles.

SEM: Examining different trimers (Au-Au-Au, Au-Ag-Au, Au@Al₂O₃-Au-Au@Al₂O₃, and Au-TiO_x-Au) to demonstrate generality.

2) **Simulations:**

More rigorous and precise FEM simulations.

Comprehensive DFT simulations for all Raman probes.

(This aspect has been thoroughly elucidated in our discussions with Reviewer #1, therefore, we will not go into detail here)

3) **Raman Investigations:**

Advanced spectrometer: Utilization of the newly acquired Horiba Xplore spectrometer for Raman measurements at 785 nm, achieving higher sensitivity.

More data for tumor subtyping: Including a substantial dataset for Raman analysis, encompassing 1600 spectra for each of the 10 distinct tumor samples.

These modifications aim to strengthen the overall quality and depth of our work. Relative modifications have been incorporated in the revised manuscript

Q3.1 The main strength of this contribution is the nanofabrication part for obtaining such Ag-Au-Ag heterotrimers. However, with respect to SERS, there are some fundamental limitations inherent to this approach. First, as in most regular SERS applications, thiol-containing analytes are detected. This is quite boring. The current challenge in SERS is to detect other than these trivial analytes without surface-seeking groups.

Reply: We appreciate the reviewer's acknowledgment of our fabrication techniques. While we acknowledge the importance of detecting more general analytes without surface-seeking groups, it is essential to underscore that the scope of this paper focuses on another important aspect, potentially conflicting with the aforementioned point.

The primary objective of our study is centered on how to tackle the inherently heterogeneous nature of hotspots for SERS applications, which is another important task in the field of SERS. Here we would like to clarify the research scope of this paper, and offer better insights for the readers. In response to the reviewer, we will try to address the comments from the following three points: 1) why we used thiol molecules in the paper; 2) motivation and significance of this work; 3) clarification of the methodology and generality.

1) Why we used thiol molecules in the paper. Thiol-containing molecules were chosen for several reasons: 1) The core focus of our paper lies in the selective detection of specific molecules rather than a broad detection of multiple analytes in solution. Molecules with surface-seeking groups are fundamental to achieving the specific detection of target molecules. 2) The primary application of our research, which involves subtyping lung tumors, relies on the detection of 4-Aminothiophenol (4-ATP). Therefore, it was imperative to present the SERS behavior of these molecules as a prerequisite before delving into the application part. 3) These molecules serve as typical Raman probes, allowing us to showcase the performance of our structures directly and enable readers to make comparisons with other studies.

2) Motivation and significance of this work. The intrinsically heterogeneous distribution of hotspots represents a fundamental challenge that significantly impedes the progress of SERS. This issue is universally acknowledged as a bottleneck in the field. In a pivotal review article authored by Luis M. Liz-Marzán and fifty other prominent researchers titled 'Present and Future of Surface-Enhanced Raman Scattering' [ACS Nano 14, 28-117 (2020)], the authors highlight this fundamental limitation in SERS development:

'The plasmonic SERS effect requires molecules to be adsorbed on surface regions of strong localized fields, that is, hotspots. The electric field distribution around a hotspot is not homogeneous and is highly dependent on the local geometric characteristics of the metallic

nanofeatures....;

Similar viewpoints have been consistently highlighted in other reviews, as evident in the work by Ren et al. [Chem. Rev. 118, 4946 (2018)], which emphasizes that:

‘However, as the enhancement is limited to a very small volume of the gap of two nanoparticles, it is more important and also challenging to bring the molecule of interest to the maximally enhanced region (hot spot) ...’

These insights underscore the significance of addressing the heterogeneous distribution of hotspots in SERS.

3) Clarification of the methodology and generality. In the revised manuscript, we introduced Au@Al₂O₃-Au-Au@Al₂O₃ trimers with smaller gaps (~2 nm). While we acknowledge that bowtie-like trimer structures have been demonstrated in the literature for applications such as photovoltaics and upconversion using EBL techniques, our proposed trimer illustrates the first time that such trimer arrays have been employed in SERS studies.

This distinction arises from the fact that the fabrication of the such trimers with EBL fundamentally requires a two-step exposure procedure, necessitating ultrahigh resolution and complicated processes, making large-area and mass production impractical. In contrast, in this paper, we utilized AAO membranes with a precise thickness of 150 ± 10 nm. The ultrathin characteristics and controllable thickness of the membrane enable the fabrication of such trimers using angle-resolved shadow deposition. It enables the production of trimers with ultrasmall gaps of ~2 nm while maintaining a large area. Even with EBL techniques, achieving such a small gap of 2 nm remains a challenge. These advantages allow us to produce sufficient and high-performance trimer samples for feasible SERS investigations.

Besides, we have also fabricated different trimers to demonstrate the generality of this work, as will be discussed in the subsequent response to **Q3.3**.

Q3.2 The presented enhancement of the structure is questionable considering the relatively large volume of the hot spot around the central gold nanostructure. The accompanying FEM simulations were performed on an object for which the dimensions are not specified, i.e., they might not correspond to the experimental situation.

Reply: We appreciate the reviewer's feedback, and in response, we have conducted rigorous revisions to enhance the manuscript. In the revised manuscript, we presented trimers with a smaller trap particle and reduced gap size of ~ 2 nm, accompanied by precise TEM characterizations as presented in **Fig. R10**:

Fig. R10 Characterizations of the $\text{Au}@Al_2O_3\text{-Au-Au}@Al_2O_3$ trimers (extracted from Fig 2).

As to the FEM simulations, we have revised this part according to the statistical results from TEM measurements as illustrated in the revised manuscript (Method Section):

'The FEM simulation parameters were set based on the following experimental parameters: According to the TEM analytical results, the long and short axes of the $\text{Au}@Al_2O_3$ ellipsoid are 43 nm and 34 nm, respectively, with a statistical error of <2 nm. The long and short axes of the trap particle are about 18 and 16 nm, respectively, with a statistical error of about 1 nm. The thickness of the Al_2O_3 ALD coating is $1.5 \text{ nm} \pm 0.1 \text{ nm}$. The gap between two adjacent particles is approximately 2 nm. The lattice of the AAO pores is 100 nm, which is rigorously determined by the anodization voltage [Nanotechnology 28, 105301(2017)].'

Relative information in this chat has been added in the revised manuscript.

Q3.3 In the SERS application for lung tumor diagnosis 4-ATP-functionalized gold nanostructure for capturing aldehyds in order to form the corresponding Schiff bases/imines (C=N) are employed. Thus, in principle any aldehyde and not only benzaldehyde (BA) claimed in this study could be detected. In other words, there is no inherent selectivity in this approach.

Reply: We appreciate the reviewer's clarification and acknowledge the error in our expressions. Here we will first clarify the confusion regarding the BA molecules. Subsequently, we will discuss the potential concerns related to selectivity and generality that the reviewer may have, with the objective of improving the clarity and rigor of the article.

First, concerning the claim about detecting only benzaldehyde (BA), we acknowledge the misunderstanding. We would like to clarify that the production of aldehydes in tumor cells involves more than just benzaldehyde. We have modified the discussions to make it clear:

For lung cancer patients, the balances of oxygen free radical breakdown, leading to the lipid peroxidation of polyunsaturated fatty acids in cancer cell membranes and organelle membranes, and consequently extra production of aldehydes and other volatile organic compounds (VOCs).⁵⁵⁻⁵⁷ Relative information has been added in the revised manuscript (Page 12).

Second, the reviewer may doubt the selectivity of our trimer configuration. We acknowledge the deficiencies in our original manuscript, particularly in terms of the trimer's ability to enrich molecules based on differences in bonding affinities. In response, we have replaced the original Ag-Au-Ag trimer configuration with the Au@Al₂O₃-Au-Au@Al₂O₃ trimers in the revised manuscript. The substantial differences between alumina and gold result in the preferential adsorption of thiol molecules on the gold trap particle. We have included new characterizations, SERS measurements, FEM and DFT simulations, and biological applications (as outlined in the '**Reply to overall comment**') to reinforce and substantiate our viewpoint. (This part has also been explained in a comprehensive manner in our response to Reviewer #2, Q2.2)

Third, addressing the generality of our approach in specific SERS detection. In the revised manuscript, we expanded our investigation to include various trimer configurations despite Au@Al₂O₃-Au-Au@Al₂O₃, including Au-Au-Au, Au-Ag-Au and Au-TiO_x-Au to showcase the versatility of our approach, as shown in **Figure R11** (Supplementary Fig. 3). The results indicate the potential for extending specific SERS detection to a broader range of conditions.

Fig. R11 Schematic diagrams, SEM images and corresponding TEM element mappings of the Au-Au-Au, Au-Ag-Au and Au-TiO_x-Au trimers, respectively (Supplementary Fig. 3).

Regarding the incorporation of oxides in trimers, it's noteworthy that thiol tends to adsorb on metals, while carboxyl, silanes, and certain other groups prefer oxide surfaces, as illustrated in **Figure R12**. Therefore, the addition of oxides in trimers can broaden the range of molecules applicable for SERS. For instance, the Au-TiO_x-Au configuration could potentially be employed to selectively enhance molecules containing carboxyl groups. However, we would like to clarify that, as researchers in plasmonics and nanofabrication, we are not experts in chemical synthesis or proficient in utilizing chemical linkers. Unfortunately, delving into the optimization of these materials and conducting corresponding SERS measurements falls outside the scope of our current study. We appreciate the valuable suggestion and acknowledge the potential for further exploration in the realm of analytical chemistry.

Fig. R12. SERS spectra of 4-Mpy and benzoic acid, respectively, on the monomer Au NPs arrays and Au@Al₂O₃ NPs arrays. The differences in SERS can be explained that thiols adsorb on gold while carboxyl adsorbed on oxides. (The monomer nanoparticle arrays were fabricated simply by a direct vertical E-beam evaporation with the AAO mask.)

The relative discussions in this chat have been incorporated in the revised manuscript and supporting information.

Q3.4: Overall, several experimental and theoretical details are missing, which makes it hard to believe the presented claims. Since this work is not addressing the broad and heterogeneous readership of Nature Communications, but contains a really nice nanofabrication part (the SERS applications lack originality and novelty), we recommend sending this contribution to a journal devoted to nanoscience or material science.

Reply: To conclude, we appreciate the constructive comments provided by the reviewer. In response to the feedback, we have implemented substantial revisions to enhance the overall quality and credibility of the manuscript. Notably, we replaced the original Ag-Au-Ag trimer configuration with the Au@Al₂O₃-Au-Au@Al₂O₃ trimers, and have included improvements in sample characterizations, SERS measurements, FEM and DFT simulations, and biological applications:

- 1) The introduction of the Au@Al₂O₃-Au-Au@Al₂O₃ trimer configuration directly addresses concerns related to selective adsorption and precise nanofabrication. Additionally, the alumina ALD shell efficiently separates adjacent metallic particles, enabling the fabrication of trimers with smaller gap distances (~ 2 nm).
- 2) To substantiate these changes, we have incorporated precise TEM characterizations of the samples and comprehensive DFT simulations into the revised manuscript. Furthermore, different trimer configurations, including Au-Au-Au, Au-Ag-Au, Au@Al₂O₃-Au-Au@Al₂O₃, and Au-TiO_x-Au, were prepared to illustrate the generality of our approach.
- 3) Moreover, we utilized a recently acquired spectrometer for Raman measurements, providing improved sensitivity and enabling rapid mapping. This instrument facilitated the acquisition of a substantial volume of data for biological analysis, offering additional insights into the subtyping of lung tumors. In fact, though SERS measurement of aldehyde molecules has been showcased with exhaled breath condensate from patients [Adv Funct Materials 32, 2202805 (2022)], conducting direct SERS analysis with human tissues remains challenging due to the limited quantity of aldehyde within cells. Notably, we performed direct Raman characterizations of aldehyde molecules from fresh tumor tissues, **marking the first instance of such an analysis**. Consequently, this approach allows in-depth investigations, including the potential for tumor subtyping through SERS and exploring the correlation between the tumor node metastasis (TNM) stage and SERS outcomes. The insights derived from our preliminary study are particularly noteworthy (see discussions of **Figure 5**). For details, please also refer to the responses to Review #5 and #6, Q 5.5.

We are confident that these modifications substantially improve the manuscript, and we appreciate the chance to respond to the reviewer's comments, thereby enhancing the overall quality of our work.

Reviewer: 5 # 6

Overall comment: In this manuscript, the authors investigate an approach to improve the sensitivity of surface-enhanced Raman scattering (SERS) for molecular sensing by using heterogeneous plasmonic trimers. Conventionally, SERS suffers from the fact that only molecules in electromagnetic hot spots, which constitute only a small proportion of the total molecules, contribute to the SERS signal. The proposed trimers increase the concentration of the probe molecules in these hot spots through chemical affinity, resulting in a significant improvement in SERS sensitivity. Furthermore, the authors demonstrate the potential of this technique in early detection of lung tumors by analyzing the SERS signal of benzaldehyde. The pathological diagnosis of different tumor subtypes is illustrated based on differences in SERS intensity, which indicates that the levels of benzaldehyde can be different in different tumor subtypes. While the manuscript presents interesting results and is well-written with high-quality figures, the novelty of the findings does not warrant publication in *Nature Communications*.

Reply: We appreciate the reviewer's positive remarks on our figures and results. To address concerns about the novelty of our findings, we have made substantial modifications to the paper. Specifically, we replaced the original Ag-Au-Ag trimer configuration with the Au@Al₂O₃-Au-Au@Al₂O₃ trimers, prompting a comprehensive revision, including improvements in sample characterizations, SERS measurements, FEM and DFT simulations, and biological applications. We would like to emphasize the novelty of our work in three key aspects:

1) Motivation. Our paper addresses the intrinsically heterogeneous distribution of hotspots, a fundamental challenge in SERS. The heterogeneity results in limited proportion of the detectable molecules among the total, leading to strong signal fluctuation that inhibits quantitative SERS analysis and also decreased efficiency in qualitative SERS. This issue is widely recognized as a bottleneck in the field, as highlighted in a pivotal review article authored by Luis M. Liz-Marzán et al. titled 'Present and Future of Surface-Enhanced Raman Scattering' [*ACS Nano* 14, 28-117 (2020)]

2) Nanofabrication: The introduction of Au@Al₂O₃-Au-Au@Al₂O₃ trimers with ultrasmall gaps (~2 nm) represents a significant advancement. Our method enables the production of trimers with ultrasmall gaps of ~2 nm while maintaining a large area. It allows us to produce sufficient and high-performance trimer samples for feasible SERS investigations. As far as we know, it is the first time that such a trimer structure has been employed in SERS studies. (detailed discussions please refer to **Q5.2**)

3) Biological Applications: While SERS measurement of aldehyde molecules has been showcased with exhaled breath condensate from patients, conducting direct SERS analysis with human tissues

remains challenging due to the limited quantity of aldehyde molecules within cells. Here we mark the first instance of direct Raman characterizations of aldehydes from fresh tumor tissues. This capability opens avenues for in-depth investigations, such as exploration of the relationship between tumor node metastasis (TNM) stage and SERS outcomes, and the investigation of tumor subtyping possibilities through SERS. (Details see **Q5.5**).

Besides, we have also fabricated different types of trimers with regard to address the concerns about the generality of approach (**Q5.4**). We believe that these modifications significantly enhance the novelty and contribution of our work, and we appreciate the opportunity to clarify these aspects based on the reviewer's feedback.

Q5.1 : There has already been extensive work investigating the concentration of target molecules in electromagnetic hotspots, such as in the following two references: **#1**. Victoria M. Szlag, Rebeca S. Rodriguez, Jiayi He, et al. *ACS Applied Materials & Interfaces* 10 (38), 31825-31844 (2018); **#2**. De Angelis, F., Gentile, F., Mecarini, F. et al. *Nature Photonics* 5, 682–687 (2011).

Reply: The reviewer referred to different strategies for molecular enrichment, including the use of affinity agents **[#1]** and the construction of super-hydrophobic structures **[#2]**. In this study, we employed a different approach by inducing molecules into hotspot regions **[#0]**. These three strategies fundamentally belong to different research directions, though all of them can be used to improve the SERS performance. **Table R1** summarizes the differences between these strategies.

Reference Number	Nanostructures (Strategies)	Strategies
#0 (this paper)		Introducing probe molecules to the 'hot spot' position
#1		Usage of affinity agents
#2		Constructing super-hydrophobic structures

Table R1 Comparison between the molecular enrichment methods listed by the reviewer. Reproduced with permission from {ACS Applied Materials & Interfaces 10, 31825 (2018)}. Copyright {(2018)} {American Chemical Society} and {Nature Photonics 5, 682 (2011)}. Copyright {(2011)} {Springer Nature}.

The first two primarily focus on macroscopic aspects, whereas our strategy emphasizes the nanoscale. Importantly, these methods are not mutually exclusive and can be employed simultaneously. For instance, we utilized affinity agents in the detection of aldehyde molecules **[#1]**, and we can also, in principle, use the drop-casting methods **[#2]**, which can potentially elevate the detection limit by two orders of magnitude or more, even on our plasmonic nanoarrays with a hydrophobic angle of 70 - 90° due to the coffee ring effects. However, drop-casting method a time-consuming process, and it may not necessarily reflect the sample's true performance. Instead, single-molecule detection is a more robust indicator of sample performance. Additionally, drop-casting methods often lead to the multilayer adsorption of molecules, which is not conducive to exploring single-molecule analysis.

We have cited the latest literature addressing SERS improvement through molecular enrichment

in the revised manuscript. While we acknowledge the importance of this issue for the broader field of SERS, it is not the primary focus of our paper.

Q5.2 : Similarly the use of bowtie-like structures to generate strong hot spots is well explored, see for example the following reference: **#3**. For photothermal applications: Pratiksha D. Dongare, Yage Zhao, David Renard, Jian Yang, Oara Neumann, Jordin Metz, Lin Yuan, Alessandro Alabastri, Peter Nordlander, and Naomi J. Halas ACS Nano 15 (5), 8761-8769 (2021); **#4**. For spectroscopy: M. Kaniber, K. Schraml, A. Regler, J. Bartl, G. Glashagen, F. Flassig, J. Wierzbowski & J. J. Finley, Scientific Reports volume 6, Article number: 23203 (2016); **#5**. For lasing: Jae Yong Suh, Chul Hoon Kim, Wei Zhou, Mark D. Huntington, Dick. Co, Michael R. Wasielewski, and Teri W. Odom, Nano Lett. 2012, 12, 11, 5769–5774

Reply: We have thoroughly reviewed the references and acknowledge the exceptional plasmonic effects observed at the sharp edges or gaps in bowtie structures. However, I would like to emphasize that, from a nanotechnology perspective, our trimer structure represents a distinct advancement in technology, and differs significantly from these dimer structures from ref **#4** and **#5** (see **Figure R13**)

Fig. R13 SEM images extracted from the provided references: ref **#4** (left) and ref **#5** (right). Reproduced with permission from {Scientific Reports 6, 23203 (2016)}. Copyright {(2016)} {Springer Nature} (left), and {Nano Lett. 12, 5769 (2012)}. Copyright {(2012)} {American Chemical Society}(right).

Regarding Ref **#3**, which presents bowtie-like trimers, we acknowledge that these structures fabricated by EBL have been demonstrated for applications such as photovoltaics and upconversion. However, our approach offers distinct advantages. It is the first time that such a trimer structure has been employed in SERS studies, the details are as follows:

The fabrication of these bowtie-like trimers with EBL typically requires a two-step exposure procedure, demanding ultrahigh resolution and complicated processes, making large-area and mass production for SERS investigations impractical. In contrast, our methodology utilizes AAO membranes with a precise thickness of 150 nm. The ultrathin characteristics and controllable thickness of the membrane enable the fabrication of such trimers using angle-resolved shadow deposition. This approach allows for the production of trimers with ultrasmall gaps of approximately

2 nm while maintaining a large area. Even with EBL techniques, achieving such a small gap remains a significant challenge. These might be the reasons that there are no available references utilizing the trimer configurations for SERS applications. Our approach holds several advantages, including 1) ultrasmall gaps, 2) large-area fabrication, 3) free from surface chemicals, 4) aligned orientations, and 5) adjustable components. These unique features distinguish our approach from comparable techniques like EBL and colloidal methods.

Q5.3 : Even plasmonic trimers themselves have been previously reported for Raman scattering:

#6 For Raman scattering: Shuangmei Zhu, Chunzhen Fan³, Erjun Liang, Pei Ding, Xiguang Dong, Haoshan Hao, Hongwei Hou & Yuanda Wu, *Scientific Reports* volume 11, Article number: 1230 (2021); **#7** For nonlinear optics: Seyfollah Toroghi, Chatdanai Lumdee, and Pieter G. Kik, *Appl. Phys. Lett.* 106, 103102 (2015);

Reply: I would like to clarify that the references mentioned (**#6** and **#7**) pertain to numerical simulations, which differ inherently from experimental research. I appreciate the reviewer's input and would like to clarify that a direct comparison between numerical simulations and experimental results may not be entirely appropriate. To know knowledge, there are no available references utilizing these trimer configurations for SERS applications.

Q5.4 : Furthermore, it is not clear how general this approach is. What if the target molecules do not have a high chemical affinity for gold? Replacing active functionalization with ligands with the intrinsic chemical affinity of the particles is not a sufficiently novel or general approach.

Reply: Thanks for your comments. We have taken them into careful consideration and modified our manuscript accordingly. In the revised manuscript, we expanded our investigation to include various trimer configurations including Au@Al₂O₃-Au-Au@Al₂O₃, Au-Au-Au, Au-Ag-Au and Au-TiO_x-Au, as depicted in **Figure R14** (also see revised Figure 2 and Supplementary Fig. 3). These additional configurations were included to highlight the versatility of our approach, demonstrating its potential for extending specific SERS detection to a broader range of conditions. We hope these enhancements contribute to the overall strength and applicability of our work.

Fig. R14 Schematic diagrams, SEM images and corresponding element mappings of the Au-Au-Au, Au-Ag-Au and Au-TiO_x-Au trimers, respectively (Supplementary Fig. 3). Please refer to revised Fig. 2 for the characterizations of the Au@Al₂O₃-Au-Au@Al₂O₃ trimers.

Q5.5 : Also the impact on lung tumor diagnosis is not clear: how does that compare with other techniques? In line 274, the authors state: “it was found that the identification of lung cancer with human breath is not always successful even for the patients which have been confirmed with lung tumors”. How the presented technique advances the state of the art for detection?

Reply: We are sorry for the confusion and we have added discussions to the revised manuscript to make it clear. In fact, SERS measurement of aldehyde molecules has been showcased with exhaled breath condensate from patients [Adv Funct Materials 32, 2202805 (2022)]. However, conducting direct SERS analysis with human tissues remains challenging due to the limited quantity of aldehyde within cells. As a result, in-depth investigations, including methods for assessing the accuracy of SERS results with tissue biopsy data, exploring the relationship between tumor node metastasis (TNM) stage and SERS outcomes, and investigating the possibility of tumor subtyping by SERS, remains to be answered.

Benefiting from the ultrahigh sensitivity of the trimer structures, we conducted direct Raman characterizations of aldehyde molecules from fresh tumor tissues, marking the first instance of such an analysis. While our sample size is limited, comprising 10 samples, the insights gained from this preliminary study are noteworthy. Our observations include:

- 1) Distinct SERS detection for different subtypes: SERS is sensitive to adenocarcinomas (ad-ca) tumors, but not sq-ca tumors. Additionally, aldehyde molecules were not detected in benign tumors neither.
- 2) Correlation with the TNM stage: Our preliminary finding suggests that there is no correlation between the SERS outcomes and the original tumor size or lymph node. For sq-ca patients,

SERS diagnosis is applicable to all the patients with TNM stages from IA1 to IIB.

We have modified this part to improve the logic and for better express our findings and significances (revised Figure 5, Page 13).

Q5.6 : From a technical point of view, the authors could also provide further details in certain places. For example, in lines 191-195, the authors state that the observed SERS enhancement of the Ag-Au-Ag trimer should not arise from the coupling between plasmonic modes and the laser wavelength. In order to show this, the field enhancement between the Ag-Au-Ag and Ag-Ag-Ag trimers should be compared. However, the authors only show the field enhancement Ag-Au-Ag trimers in Figure 3b, while in Supplementary Figure 4, they show only the extinction spectra and not the field enhancement.

Reply: The comparisons have been added in the revised files. Considering that we used the Au@Al₂O₃-Au-Au@Al₂O₃ trimers in the revised manuscript, and Au-Au-Au trimers as a comparison, all the corresponding simulations have been replaced accordingly.

Relative revisions have been included (Revised Fig.2 and Supplementary Fig. 5)

REVIEWER COMMENTS

Reviewer #1 (Remarks to the Author):

To my satisfaction, the authors explained, corrected, and implemented all concerns raised (by me) in the first round. In the current improved state, I suggest the manuscript for publication in Nature Communications.

Reviewer #2 only provided confidential Remarks to the Editor.

Reviewer #3 (Remarks to the Author):

Reply to rebuttal:

The authors have included an extensive rebuttal and modified the manuscript to a large extent.

The authors essentially have complained about my criticism regarding the thiol-containing analyte. Unfortunately, they have not included new experimental results with other, non-thiol-containing analytes. In summary, a major concern has not been addressed satisfactorily. The value of SERS applications is largely based on the analytes that are detectable by it and thus, analytes are at the very heart of every SERS application, contrary to the authors' opinion. To be more clear: A strong hot spot that is only useful for "trivial analytes" is of little use.

I am still of the opinion that the presented nanofabrication part alone does not provide sufficient novelty and originality to justify publication in Nature Communications. It must be connected with some new physical or chemical insight.

Surprisingly, the authors state that a justification of the benefits of using such a plasmonic trimer is beyond the scope of their contribution. On the contrary, I think that the justification of preparing such a plasmonic trimer must be the main motivation before undertaking such a study. In other words: What do we learn from such a structure that we did not know before? From a very basic scientific perspective: every scientific publication should address a clear research question.

The nanofabrication of the plasmonic trimer is only relevant to the broad and heterogeneous readership of Nature Communications if the reader learns something from it. However, this is not the case here.

Therefore, I recommend submission to a specialized journal focused on material and nano science.

Reviewer #4 (Remarks to the Author):

Reviewer #5 (Remarks to the Author):

The authors have vastly changed the original manuscript. Overall, the quality of the paper has

improved, and the scope is clearer. In the new manuscript, it is clarified that a key advantage of the proposed approach is the capability to fabricate extended nanostructured samples featuring heterogeneous trimers where the central element may display chemical properties different from the lateral structures. For example, it is shown how the thiols preferentially adsorb on Au (central element) rather than Al₂O₃ (the coating of the lateral elements). Therefore, the idea is to use these heterogeneous trimers to preferentially concentrate molecules with thiol groups on the Au central nanostructure. Such concentration, together with the field enhancement also occurring at the same location, provides an increased SERS signal.

The new manuscript could be suitable for publication, provided the authors address some comments:

- In the introduction (line 66-67), the authors discuss some drawbacks of different techniques to improve SERS sensitivity and mention: ‘...but the introduction of extra chemical species may also restrict the access of target molecules.’ However, isn’t this what they do when applying the trimers for the identification of cancerous tissues using 4-ATP-modified samples? Another reviewer brought this up, and I saw the previous answer from the authors. Even assuming the other reviewer accepts the explanation, I think the manuscript should be consistent, and the discussion in the introduction should be modified accordingly.

- Regarding the application for lung cancer: I think it would be useful to quantitatively compare the signal from Au-Au-Au and Au@Al₂O₃-Au-Al₂O₃@Au in the case of ad-ca tissues, to show the improved signal in the case where the alumina coating is present. The authors have already done that for Ag-Au-Ag in the previous version of the manuscript. It would be relevant to compare the three cases.

Reviewer #6 (Remarks to the Author):

Point-by-Point Replies to the referees: (turn #2, NCOMMS-23-09102B, 5th April 24, 2024)

Reviewer: 1

Reviewer's Comment: To my satisfaction, the authors explained, corrected, and implemented all concerns raised (by me) in the first round. In the current improved state, I suggest the manuscript for publication in Nature Communications.

Reply: We thank the reviewer for the positive feedback.

Reviewer: 2

Reviewer's Comment: Reviewer #2 only provided confidential Remarks to the Editor.

Reply: According to the comment from the editor, it is indicated that the concern might from our editorial perspective be related to previous publications with similar aldehyde detection concepts and, in part, using lung cancer samples. In response, we systematically reviewed existing literature on SERS-based detection of aldehydes, as summarized in **Table R1**. These studies can be categorized into two groups: those involving SERS measurements of aldehydes without human subjects (#1, #2, #6), and those using exhaled gases collected from individuals over an extended period (#3, #4, #5). While these studies have demonstrated the feasibility of detecting aldehyde molecules using SERS, they typically lack direct links between SERS outcomes and tumor pathological data. This is primarily due to the challenges of conducting SERS analysis with human tissues, given the limited quantity of aldehydes within cells. Consequently, further investigations, such as tumor subtyping and analysis of cancer stages, have been limited.

Table R1 Representative papers utilizing SERS for the detection of aldehydes.

Number	Methods	Paper
#1	Conceptual model excluding human subjects	Adv. Mater. 30, 1702275 (2018)
#2	Conceptual model excluding human subjects	Angew. Chem. Int. Ed. 58, 16523 (2019)
#3	Exhaled gas from human	Anal. Chem. 93, 4924 (2021)
#4	Exhaled gas from human	Adv. Funct. Mater. 32 2202805 (2022)
#5	Exhaled gas from human	Small 19, 2207324 (2023)
#6	Conceptual model combined with machine learning, excluding human subjects	ACS Sensors 8, 3487 (2023)

In this paper, benefiting from the ultrahigh sensitivity of the trimer structures, direct Raman

characterizations of aldehydes from fresh tumor tissues were conducted, marking the first instance of such an analysis. Our observations include: **1) SERS detection for tumor subtyping:** Our study reveals that SERS is sensitive to adenocarcinoma (ad-ca) tumors, while it does not exhibit significant sensitivity to squamous carcinoma (sq-ca) tumors or benign tumors. **2) Correlation SERS with the TNM stage:** Our preliminary findings suggest SERS diagnosis is applicable to all patients with TNM stages from IA1 to IIB for sq-ca patients. No clear evidence suggests that there is a correlation between the SERS outcomes and the TNM stage under quantitative measurement. The observations offer insights that have received limited attention in prior studies.

Hence, although the concept has been proposed in previous studies, our paper provides the first instance of conducting pathological SERS analysis with fresh tumor tissues, offering new insights into this field. In response to the comments, the title, abstract and main text have been modified to emphasize the specific application scenarios of our SERS substrates, as suggested by the editor (marked in red).

Reviewer: 3

Reviewer's Comment: The authors have included an extensive rebuttal and modified the manuscript to a large extent. The authors essentially have complained about my criticism regarding the thiol-containing analyte. Unfortunately, they have not included new experimental results with other, non-thiol-containing analytes. In summary, a major concern has not been addressed satisfactorily. The value of SERS applications is largely based on the analytes that are detectable by it and thus, analytes are at the very heart of every SERS application, contrary to the authors' opinion. To be more clear: A strong hot spot that is only useful for "trivial analytes" is of little use. I am still of the opinion that the presented nanofabrication part alone does not provide sufficient novelty and originality to justify publication in Nature Communications. It must be connected with some new physical or chemical insight. Surprisingly, the authors state that a justification of the benefits of using such a plasmonic trimer is beyond the scope of their contribution. On the contrary, I think that the justification of preparing such a plasmonic trimer must be the main motivation before undertaking such a study. In other words: What do we learn from such a structure that we did not know before? From a very basic scientific perspective: every scientific publication should address a clear research question.

Reply: We acknowledge the feedback provided by the reviewer. In response, we have included additional SERS measurements using non-thiol analytes, including adenine (amino), thiram (disulfide), and benzeneselenol (selenol) according to the suggestions by the editor. The rationale for the selection of these molecules and their SERS performance are discussed:

- 1) Principles for molecule selection:** Based on our literature review, we have identified several functional groups, including amines, disulfides, carboxylic acids, phosphines, and selenols that exhibit an affinity for adsorption onto gold surfaces. Among these molecules, carboxylic acids and phosphines were excluded due to their strong affinity for Al_2O_3 . Therefore, amines, disulfides and selenols were selected for further considerations. For amines, their amino groups establish strong coordinative bonds with the gold surface due to the affinity between the lone pair of electrons on the nitrogen atom and the gold atoms. Conversely, though amines can also interact with hydroxyl groups on the alumina surface, they form hydrogen bonds that exhibit a significant lower adsorption capacity [(Adv. Mater. 2211624 (2023); J. Phys. Chem. B 109, 15150 (2005); J. Phys. Chem. C 114, 10512 (2010)]. For disulfide molecules, they sharing similar characteristics with thiol analytes by establishing strong surface bonding with gold, while exhibiting inferior affinity to alumina [J. Am. Chem. Soc. 109, 733 (1987), J. Am. Chem. Soc. 105, 4481 (1983)]. For selenol molecules, they firmly attach to gold surfaces by

establishing Au-Se bonds, demonstrating a markedly enhanced affinity towards gold over alumina [Langmuir 14, 4802 (1998)]. Hence, adenine (amines), thiram (disulfides) and benzeneselenol (BSe, selenols) molecules were selected for SERS analysis.

2) SERS performance.

2-1. Preliminary Proof. Comparative SERS performance of these three non-thiol molecules on monomer Au NPs arrays and monomer Au@Al₂O₃ NPs arrays were shown in **Fig. R1**. The results in **Fig. R1** reveal that adenine (a), thiram (b) and BSe (c) exhibit better performance on the AuNPs, while no signal was observed on the Au@Al₂O₃ NPs. In contrast, benzoic acid, a molecule with a carboxy group that prefers adsorption on alumina, exhibits better SERS performance on the Au@Al₂O₃ NPs (d). The results suggest that these three analytes will preferentially be adsorbed on the Au NPs rather than the Au@Al₂O₃ NPs. (Details for the fabrication of the monomer particle arrays were available in our previous publications (Adv. Mater. 30, 1705421 (2018)))

Fig. R1. SERS spectra of adenine (a), thiram (b), BSe (c) and benzoic acid (d) on the monomer Au NPs arrays and Au@Al₂O₃ NPs arrays, respectively.

2-2. Further SERS Measurement. SERS performance of these analytes from the Au@Al₂O₃-Au-Au@Al₂O₃ and Au-Au-Au trimers was compared in **Fig. R2**. The results reveal a significant improved SERS performance for the molecule on the Au@Al₂O₃-Au-Au@Al₂O₃ trimers, especially at low molecular concentrations, indicating enhanced efficiency in enriching the molecules at the central gap position. This offers a feasible solution against the intrinsically heterogeneous distribution of ‘hot spot’, a well-recognized fundamental challenge that significantly impedes the development of SERS [ACS Nano 14, 28-117 (2020)], leading to a substantial increase in the proportion of detectable molecules and subsequent improvement in SERS sensitivity. Relative data and discussions have been supplemented in the revised manuscript (marked in red) and supplementary file (Supplementary Fig. 11)

Fig. R2 SERS spectra of adenine (**a, b**), thiram (**c, d**) and BSe (**e, f**) from the Au@Al₂O₃-Au-Au@Al₂O₃ and Au-Au-Au trimers, respectively, at different concentrations.

2-3. What can expect from this strategy in the future. In this paper, we demonstrated that molecules with a strong affinity to gold can be significantly enhanced by the Au@Al₂O₃-Au-Au@Al₂O₃ trimer structures. In fact, similar structures with reversed configurations, featuring oxides in the center and metals on the sides, can be fabricated using a similar approach, as shown in **Fig. R3**. This extension of our SERS arrays can potentially broaden the range of detectable molecules to include groups like carboxyl, catechol, and phosphate groups, which exhibit a strong affinity to oxides. However, these results, while sharing similarities with the proposed trimers, do not fit the general definition of trimers because they consist of only a thin film in the middle instead of a particle. The design differs from the trimers as the oxides in the middle cannot provide LSPRs response, necessitating a narrowed gap between the dimers. Given the logical coherence of this paper and the current progress, the results were not included in this submitted paper.

Fig. R3 was blinded here as it will be used in a following submission.

Reviewer: 4

Reviewer: 5

Overall Comment: The authors have vastly changed the original manuscript. Overall, the quality of the paper has improved, and the scope is clearer. In the new manuscript, it is clarified that a key advantage of the proposed approach is the capability to fabricate extended nanostructured samples featuring heterogeneous trimers where the central element may display chemical properties different from the lateral structures. For example, it is shown how the thiols preferentially adsorb on Au (central element) rather than Al_2O_3 (the coating of the lateral elements). Therefore, the idea is to use these heterogeneous trimers to preferentially concentrate molecules with thiol groups on the Au central nanostructure. Such concentration, together with the field enhancement also occurring at the same location, provides an increased SERS signal. The new manuscript could be suitable for publication, provided the authors address some comments:

Major Concern: Regarding the application for lung cancer: I think it would be useful to quantitatively compare the signal from Au-Au-Au and $\text{Au}@/\text{Al}_2\text{O}_3\text{-Au-Au}@/\text{Al}_2\text{O}_3$ in the case of ad-ca tissues, to show the improved signal in the case where the alumina coating is present. The authors have already done that for Ag-Au-Ag in the previous version of the manuscript. It would be relevant to compare the three cases.

Reply: Thanks for your positive comments. We have incorporated additional experiments in accordance with the suggestions provided. The Raman mappings of an ad-ca sample (patient #11, pT1aN0M0, IA2, 1.8*1.6*1.0 cm) from the $\text{Au}@/\text{Al}_2\text{O}_3\text{-Au-Au}@/\text{Al}_2\text{O}_3$ and Au-Au-Au trimers were compared in **Fig. R3**. Relative data has been included in Supplementary Fig. 19. Accordingly, data for patient #11 has been updated in Supplementary Table 1.

Fig. R3. Raman mapping at 1623 cm^{-1} of an ad-ca sample (patients #11) from the $\text{Au}@/\text{Al}_2\text{O}_3\text{-Au-Au}@/\text{Al}_2\text{O}_3$ (a) and Au-Au-Au (b) trimer arrays. Both samples had been functionalized with 4-ATP molecules, and immersed into the solution containing target tissue simultaneously during the preparations.

On the other hand, the data regarding the Ag-Au-Ag trimers presented during our first submission was not included here due to the following reasons: 1) The laser conditions were adjusted from 1 mW, 633 nm to 0.5 mW, 785 nm to match the LSPR modes of Au. 2) The map data presented in this revised manuscript was obtained through direct Raman mapping from a single sample, whereas the

map data for the Ag-Au-Ag trimers was based on typical spectra from different areas of four different samples, selected from a larger collection of measurement points. Therefore, a direct comparison between the Ag and Au trimers may not offer meaningful insights for readers. As a result, we made the decision to omit the data on Ag-Au-Ag trimers in the revised manuscript.

Minor concern: In the introduction (line 66-67), the authors discuss some drawbacks of different techniques to improve SERS sensitivity and mention: ‘...but the introduction of extra chemical species may also restrict the access of target molecules.’ However, isn’t this what they do when applying the trimers for the identification of cancerous tissues using 4-ATP-modified samples? Another reviewer brought this up, and I saw the previous answer from the authors. Even assuming the other reviewer accepts the explanation, I think the manuscript should be consistent, and the discussion in the introduction should be modified accordingly.

Reply: We have modified our expressions according to the suggestion to maintain the consistency throughout the manuscript.

Reviewer: 6

REVIEWER COMMENTS

Reviewer #3 (Remarks to the Author):

The authors have revised the manuscript to a significant extent and changed the perspective to a "plasmonic trap". The central bare gold particle acts as the "chemical trap" for attracting the analytes with surface-seeking groups. The Al₂O₃-passivated large gold nanoparticles left and right contribute to plasmonic coupling in this trimer and provide the necessary SERS enhancement. This material science-oriented story alone is convincing, but not sufficiently novel and original for justifying publication in Nature Communications.

My suggestion to use also non-thiol-based analytes in the SERS applications originally aimed at using analytes that do not have a surface-seeking group for strong chemisorption. In contrast, the authors have now replace the thiol moiety as a surface-seeking group by other surface-seeking groups with similar affinity to gold: disulfide Se instead S and amino groups. So the applicability to a large set of molecules with various surface-seeking groups has been demonstrated, but the actual demonstration of using weakly adsorbing analytes without surface-seeking has not been demonstrated. Most importantly, for toluene as "non-trivial" analyte (Fig. 3b) one clearly sees that the concept does not work out since toluene has a similar affinity to both gold and the aluminum oxide shell. Thus, the concept of the "plasmonic" traps only seems to work for "trivial" analytes that have high affinity to gold. In other words, it is not generally applicable but only works for a rather narrow range of compounds with strong surface-seeking groups.

Overall, my estimation of the scientific value of this contribution has not substantially changed: the material science part is nice, the SERS part is too ordinary. Therefore, I still recommend submission to a specialized journal focused on material and nano science.

Reviewer #4 (Remarks to the Author):

Reviewer #5 (Remarks to the Author):

The authors have addressed my concerns. I can suggest the publication provided that also the other reviewers' comments are addressed as well.

Reviewer #6 (Remarks to the Author):

Reviewer #7 (Remarks to the Author):

This manuscript introduces an innovative approach to tackle the challenge of heterogeneous hot spot distribution in plasmonic nanostructures, a crucial aspect in SERS research. The authors present the fabrication of Au@Al₂O₃-Au-Au@Al₂O₃ trimers, consisting of paired core-shell dimers with a trap

plasmonic particle in between. This configuration drives probe molecules towards central traps via chemical affinity, ensuring precise spatial alignment between probes and hot spots, thus significantly augmenting SERS sensitivity. Experimental findings illustrate remarkable enhancements in SERS sensitivity across various molecules, achieving single-molecule detection. Furthermore, the manuscript delves into the potential of SERS for early detection of lung tumors by analyzing aldehyde signals extracted from fresh tumor tissues.

The manuscript offers compelling evidence of enhanced SERS sensitivity through the proposed Au@Al₂O₃-Au-Au@Al₂O₃ trimers and their potential for early tumor detection. However, despite undergoing a thorough examination in the first round of reviewing, critical issues remain that require clarification. Specifically, the authors must address questions 3 and 4 in greater detail.

1) In Figures S1 and S2, the authors illustrate the controllability of the gap distance between Au@Al₂O₃ dimers and the size of the central Au particle. It is imperative for the authors to elucidate the reasoning behind optimizing the proposed Au@Al₂O₃-Au-Au@Al₂O₃ trimers showcased in Figures 3-5.

2) The manuscript should discuss the potential impact of varying the thickness of the Al₂O₃ layer on the SERS properties.

3) Figures 2i and S5c depict E-field enhancement values for Au@Al₂O₃-Au-Au@Al₂O₃ ($E/E_0=87.2$) and Au-Au-Au trimers ($E/E_0=43.1$). However, it is noted that introducing a dielectric layer into the plasmonic nanogap typically diminishes the overall E-field enhancement. The authors are prompted to address this disparity between simulation data and established knowledge. One plausible explanation lies in the reduced gap distance due to the thin Al₂O₃ layer. This thin layer may effectively behave like Au, thus narrowing the gap distance and allowing for surface plasmon coupling to occur. This phenomenon is supported by the E-field simulation presented in Fig. 2i.

4) In Figure 3, the authors attribute the heightened sensing capability of Au@Al₂O₃-Au-Au@Al₂O₃ trimers to the trapping effect of the core Au particles, facilitated by the strong chemical affinity of thiols. However, the stronger E-field generation of Au@Al₂O₃-Au-Au@Al₂O₃ trimers compared to Au-Au-Au trimers suggests that the observed enhancement stems from the increased E-field rather than selective adsorption of analytes onto the central Au island. To resolve this discrepancy, synthesizing Au-Au-Au trimers with identical physical dimensions as Au@Al₂O₃-Au-Au@Al₂O₃ and conducting SERS measurements on both substrates would be necessary. This comparative analysis would elucidate whether the observed lower limit of detection arises from E-field enhancement or the concentration of adsorption on the central Au island.

5) The authors are encouraged to provide references concerning the chemical reaction of 4-ATP with benzaldehyde molecules. Furthermore, it would be beneficial to suggest references or conduct Density Functional Theory (DFT) calculations supporting the origin of the peak at 1623 cm⁻¹. This peak should be confirmed experimentally through SERS measurements of the same chemicals with -C=N- bonding, directly adsorbed onto the substrates, rather than through surface reactions.

Point-by-Point Replies to the referees: (turn #3, NCOMMS-23-09102C, 26th May 24, 2024)

Reviewer #3 & #4(Remarks to the Author):

The authors have revised the manuscript to a significant extent and changed the perspective to a "plasmonic trap". The central bare gold particle acts as the "chemical trap" for attracting the analytes with surface-seeking groups. The Al₂O₃-passivated large gold nanoparticles left and right contribute to plasmonic coupling in this trimer and provide the necessary SERS enhancement. This material science-oriented story alone is convincing, but not sufficiently novel and original for justifying publication in Nature Communications. My suggestion to use also non-thiol-based analytes in the SERS applications originally aimed at using analytes that do not have a surface-seeking group for strong chemisorption. In contrast, the authors have now replace the thiol moiety as a surface-seeking group by other surface-seeking groups with similar affinity to gold: disulfide Se instead S and amino groups. So the applicability to a large set of molecules with various surface-seeking groups has been demonstrated, but the actual demonstration of using weakly adsorbing analytes without surface-seeking has not been demonstrated. **Most importantly, for toluene as "non-trivial" analyte (Fig. 3b) one clearly sees that the concept does not work out since toluene has a similar affinity to both gold and the aluminum oxide shell. Thus, the concept of the "plasmonic" traps only seems to work for "trivial" analytes that have high affinity to gold. In other words, it is not generally applicable but only works for a rather narrow range of compounds with strong surface-seeking groups.** Overall, my estimation of the scientific value of this contribution has not substantially changed: the material science part is nice, the SERS part is too ordinary. Therefore, I still recommend submission to a specialized journal focused on material and nano science.

Reply: Our study specifically focuses on addressing the heterogeneous distribution of hot spots. It is known that the hot spots only account for 1% of the surface area, which strongly limits the potential of SERS for ultrasensitive detection. This is a commonly recognized challenge within the SERS community, as detailed in the introduction of the manuscript. To address this challenge, a specifically designed oxide-metal-oxide trimer configuration was proposed in this study. The trimer configuration targets analytes to the central metals where hot spots are located, ensuring the overlap between adsorption sites and hot spots, offering a feasible route against the heterogeneous hot spots. As we can see, this approach fundamentally relies on the chemical affinity of probe molecules. However, usage of weakly adsorbing analytes generally refers to a situation that molecules are randomly adsorbed on the surface, and consequently leading to a fact that only a small portion of molecules can be located in the hot spots. As we can see, there is a conflict between the two cases.

Our approach aims to provide a general method for the fabrication of trimer arrays. In this article,

we presented different trimers, including Au@Al₂O₃-Au-Au@Al₂O₃, Au-Au-Au, Au-Ag-Au and Au-TiO_x-Au. The functionality fundamentally relies on the chemical affinity between the central gold particle and the probe molecules. Therefore, molecules with strong affinity to gold were employed. However, it does not necessarily mean that this approach is limited to this specific situation. To demonstrate the versatility of our approach, the **metal-oxide-metal** trimer was presented, as shown in **Fig. R1**. Here TiO_x was deposited at the gap position between silver dimers to trap molecules into the hot spots. By incorporating oxide as the central trap material instead of metal, the range of detectable molecules was broadened to hydroxy, carboxyl, and other groups that typically adsorb on oxides, as depicted in **Fig. R2**.

Fig. R1 and Fig. R2 were blinded here as they will be used in the following publications.

This trimer was presented to demonstrate the versatility of our approach. Considering TiO_x does not have a response in LSPR, the distance between the bilateral plasmonic particles was changed accordingly, resulting in a different morphology compared to the configuration used in the manuscript. Given the extensive data and logical coherence presented in this paper, we intend to publish them in a follow-up work. Besides, we have also conducted SERS measurement of several 'weak adsorbing molecules' like dopamine, thymine, using the trimers presented in the manuscript. As expected, the SERS performance of these molecules is nice, but show no significant differences with other plasmonic structures with sharp gaps, such as dimers [ACS Appl. Mater. Interfaces 14, 54174 (2020)]. As a result, the unique significance and advantages of the trimers cannot be highlighted, and thus were not included in the manuscript.

Reviewer #7 (Remarks to the Author):

This manuscript introduces an innovative approach to tackle the challenge of heterogeneous hot spot distribution in plasmonic nanostructures, a crucial aspect in SERS research. The authors present the fabrication of Au@Al₂O₃-Au-Au@Al₂O₃ trimers, consisting of paired core-shell dimers with a trap plasmonic particle in between. This configuration drives probe molecules towards central traps via chemical affinity, ensuring precise spatial alignment between probes and hot spots, thus significantly augmenting SERS sensitivity. Experimental findings illustrate remarkable enhancements in SERS sensitivity across various molecules, achieving single-molecule detection. Furthermore, the manuscript delves into the potential of SERS for early detection of lung tumors by analyzing aldehyde signals extracted from fresh tumor tissues. The manuscript offers compelling evidence of enhanced SERS sensitivity through the proposed Au@Al₂O₃-Au-Au@Al₂O₃ trimers and their potential for early tumor detection. However, despite undergoing a thorough examination

in the first round of reviewing, critical issues remain that require clarification. Specifically, the authors must address questions 3 and 4 in greater detail.

Reply: Thanks for your positive feedback. Here is point-by-point reply to the comments.

Q7.1: In Figures S1 and S2, the authors illustrate the controllability of the gap distance between Au@Al₂O₃ dimers and the size of the central Au particle. It is imperative for the authors to elucidate the reasoning behind optimizing the proposed Au@Al₂O₃-Au-Au@Al₂O₃ trimers showcased in Figures 3-5.

Reply: In the manuscript, we employed the trimer in Fig. S2(c) for the following experiments because it offers a high proportion of hotspots relative to the total adsorption sites. The details are explained as following:

Given that molecules may adsorb on the open surface of non-hotspot areas of the trap particle (Raised by Reviewer #2 in the first round), a smaller trap can facilitate the molecular enrichment at hot spots. As illustrated in **Fig. R3**, assuming that sites with E/E_0 higher than 20 ($EF = 1.6 \times 10^5$) can be considered as hot spots, the probability of a site being a hotspot is 18.1% for this Au@Al₂O₃ trimers ($S_{\text{traphot}}/S_{\text{trap}}$). This probability decreases with the trap diameter, namely that a smaller trap nanoparticle corresponds to a higher percentage and subsequently improved SERS performance. Therefore, we employed the configuration in Fig. S2c, with a trap particle diameter of ~10 nm, for our experiments. Such a dimension is close to the resolution limit of SEM, and consequently the limit of our approach. We have modified relative discussions to clarify the concern. (Figure S2 in the Supplementary Information).

Fig. R3 3D waterfall map representing the EM intensity around the trimers.

Q7.2: The manuscript should discuss the potential impact of varying the thickness of the Al₂O₃ layer on the SERS properties.

Reply: Thanks for your suggestion. In this study, the thickness of the Al₂O₃ layer was 1.5 nm, this parameter was chosen out of a balance of the shape retention effects in ALD and steric hindrance effects. The details are explained as following:

The Al₂O₃ layer was controlled by ALD, characterized by 0.1 nm per cycles. ALD requires a

deposition of at least of approximately 10 cycles (~ 1 nm) to form dense and continuous film for the shape retention [Nature 464, 392 (2010)]. In this case, molecules preferentially adsorb on the high-curvature surfaces of nanoparticles, as illustrated in **Fig. R4** [Langmuir 39, 15828 (2023)]. However, increase in ALD thickness, namely decrease in gap distance, will break the isotherm, namely more molecules will be adsorbed on the non-hotspots area. Moreover, when the gap size is comparable to the molecules attempting to enter, molecules physically cannot fit through the gap due to steric hindrance. The length scale is $\sim 0.5 - 0.6$ nm for 4-Mercaptobenzoic acid (4-MBA) [ACS Meas. Sci. Au 3, 434 (2023)] and $1.1 - 1.6$ nm for Rhodamine 6G [J. Chem. Phys. 112, 10435 (2000)]. Therefore, increase in ALD thickness may potentially lead to a decrease in SERS performance.

Fig. R4 Langmuir adsorption isotherms of 4-MBA on gold nanoparticles. Reproduced with permission from {Langmuir 39, 15828 (2023)}. Copyright {(2023)} {American Chemical Society}.

To support our viewpoint, we prepared $\text{Au@Al}_2\text{O}_3$ -Au-Au@ Al_2O_3 samples with different alumina shell thicknesses for SERS measurement, as shown in **Fig. R5**. It is observed that $\text{Au@Al}_2\text{O}_3$ trimers with a thickness of 1.5 nm (15 cycles) exhibited the best SERS performance. As a contrast, when 5 cycles of alumina were deposited, some gold atoms will be exposed to the environment because the alumina coating was not no continuous, weakening the “trapping effect”. Additionally, with 25 cycles of alumina coating, a decrease in SERS signals was also observed, indicating the negative size exclusion effects at play.

Fig. R5 SERS signal of 10^{-11} M 4-Mpy molecules on Au@Al₂O₃-Au-Au@Al₂O₃ trimer arrays with different alumina shell thicknesses.

In this study, we utilized a 1.5 nm ALD layer as the parameter. The addition of this layer resulted in a red-shift of 14 nm compared to bare Au trimers (Figure 2a and Figure S5a) due to the changes in dielectric environment. However, modifications in ALD thickness from 1 nm to 2.5 nm only induced very slight changes in LSPR response, which were not discernible by our UV-vis spectrometer, indicating a limited impact on SERS. Considering the shape retention effects in ALD and convenience for TEM characterizations, we opted for the 1.5 nm ALD thickness (Relative data and discussions have been included in the revised files, Supplementary Fig. 8).

Q7.3: Figures 2i and S5c depict E-field enhancement values for Au@Al₂O₃-Au-Au@Al₂O₃ ($E/E_0=87.2$) and Au-Au-Au trimers ($E/E_0=43.1$). However, it is noted that introducing a dielectric layer into the plasmonic nanogap typically diminishes the overall E-field enhancement. The authors are prompted to address this disparity between simulation data and established knowledge. One plausible explanation lies in the reduced gap distance due to the thin Al₂O₃ layer. This thin layer may effectively behave like Au, thus narrowing the gap distance and allowing for surface plasmon coupling to occur. This phenomenon is supported by the E-field simulation presented in Fig. 2i.

Reply: Thanks for your comments. This disparity between our data and established knowledge arises from several factors as following:

1) Typically, introducing a dielectric layer may diminish the E-field enhancement because the molecular adsorption site was changed. As illustrated in **Fig. R6(a)**, the adsorption site was changed from A to A' due to the additional dielectric layer, moving away from the gold surface by approximately 1-2 nm. Considering that the EM fields decrease exponentially (10^4) with distance, a significant reduction in EM intensity can be expected for the adsorbed molecules. However, in this paper, the molecular adsorption sites were B and B', as depicted **Fig. R6(b)**. Since the distance between two gold nanoparticle remains constant after the introduction of the oxide layer, the EM variations around the gap region for the two cases in Figure R4b is not that significant. This is the primary reason that why diminished E-field enhancement is not observed after the introduction of a dielectric layer.

Fig. R6 Schematics illustrating the molecular adsorption sites for conventional condition and our experimental condition.

2) Secondly, in addition to the plausible explanation raised in your comments (**Fig. R7**, high-resolution version of Fig. 2i), the introduction of the Al₂O₃ layer leads to a shift of ~14 nm in LSPR modes (Figure 2a and Figure S5a), close to wavelength of the excitation laser at 785 nm. Therefore, enhanced EM fields were observed for the Au@Al₂O₃-Au-Au@Al₂O₃ trimers in the simulations.

Fig. R7 High-resolution version of Figure 2i illustrating the field distribution around the Au@Al₂O₃-Au-Au@Al₂O₃ trimers.

Q7.4: In Figure 3, the authors attribute the heightened sensing capability of Au@Al₂O₃-Au-Au@Al₂O₃ trimers to the trapping effect of the core Au particles, facilitated by the strong chemical affinity of thiols. However, the stronger E-field generation of Al₂O₃ trimers compared to Au-Au-Au trimers suggests that the observed enhancement stems from the increased E-field rather than selective adsorption of analytes onto the central Au island. To resolve this discrepancy, synthesizing Au-Au-Au trimers with identical physical dimensions as Au@Al₂O₃-Au-Au@Al₂O₃ and conducting SERS measurements on both substrates would be necessary. This comparative analysis would elucidate whether the observed lower limit of detection arises from E-field enhancement or the concentration of adsorption on the central Au island.

Reply: Thanks for your suggestion. We prepared the Au-Au-Au trimers with a smaller gap distance, which is close to the dimension of the Au@Al₂O₃-Au-Au@Al₂O₃ trimers, and single-molecule

SERS analysis was performed to illustrate the SERS performance, as depicted in **Fig. R8**. The comparative results reveal that the Au@Al₂O₃-Au-Au@Al₂O₃ sample exhibited more positive points, suggesting that the SERS enhancement is primarily attributed to the “trapping effects”. Relative information and discussions have been included in the revised file (Supplementary Fig. 7).

Fig. R8 Raman intensity mapping of 10⁻¹¹ M 4-Mpy from the Au@Al₂O₃-Au-Au@Al₂O₃ (a) and Au-Au-Au trimer with a ultrasmall nanogap (b), respectively (integral time 500 ms). (c) HAADF-STEM image of the Au-Au-Au trimers with a ultrasmall gap.

Q7.5: The authors are encouraged to provide references concerning the chemical reaction of 4-ATP with benzaldehyde molecules. Furthermore, it would be beneficial to suggest references or conduct Density Functional Theory (DFT) calculations supporting the origin of the peak at 1623 cm⁻¹. This peak should be confirmed experimentally through SERS measurements of the same chemicals with -C=N- bonding, directly adsorbed onto the substrates, rather than through surface reactions.

Reply: Thanks for your comments. Reference concerning the reaction between 4-ATP and benzaldehyde molecules [Adv. Mater. 30, 1702275 (2018), Angew. Chem. Int. Ed. 58, 16523 (2019)], and suggesting the stretching vibration mode of C = N at 1623 cm⁻¹ [J. Raman Spectrosc. 34, 737 (2003), J. Appl. Spectrosc. 59, 5 (1993)] have been included in the revised manuscript.

REVIEWERS' COMMENTS

Reviewer #7 (Remarks to the Author):

This is a follow-up review to my previous comments on the original manuscript. As noted in my initial report, my primary concerns were outlined in questions 3 and 4. The authors have conducted additional calculations and experiments to address these concerns and have done so convincingly. Therefore, I now recommend the acceptance of this manuscript for publication in Nature Communications.